

# Quantifying Thresholds of Barrier Geomorphic Change in a Cross-Shore Sediment Partitioning Model

Daniel J. Ciarletta[1], Jennifer L. Miselis[1], Justin L. Shawler[2], Christopher J. Hein[2]

[1]U.S. Geological Survey, St. Petersburg Coastal and Marine Science Center, 600 4th St. S, St. Petersburg, Florida 33701, USA
[2]Virginia Institute of Marine Science, William & Mary, P.O. Box 1346, Gloucester Point, Virginia 23062, USA

*Correspondence to*: Daniel J. Ciarletta (dciarletta@usgs.gov)

**Abstract.** Barrier coasts, including barrier islands, beach-ridge plains, and associated landforms, can assume a broad spectrum of morphologies over decadal scales that reflect conditions of sediment availability, accommodation, and relative sea-level rise. However, the quantitative thresholds of these controls on barrier-system behavior remain largely unexplored, even as modern sea-level rise and anthropogenic modification of sediment availability increasingly reshape the world's sandy coastlines. In this study, we conceptualize barrier coasts as sediment partitioning frameworks, distributing sand delivered from the shoreface to the subaqueous and subaerial components of the coastal system. Using an idealized morphodynamic model, we explore thresholds of behavioral/morphologic change over decadal to centennial timescales, simulating barrier evolution within quasi-stratigraphic morphological cross-sections. Our results indicate a wide diversity of barrier behaviors can be explained by the balance of fluxes delivered to the beach versus the dune/backbarrier, including previously understudied forms of transgression that allow the subaerial system to continue accumulating sediment during landward migration. Most importantly, our results show that barrier state transitions between progradation, cross-shore amalgamation, aggradation, and transgression are controlled largely through balances within a narrow range of relative sea-level rise and sediment flux. This suggests that, in the face of rising sea levels, subtle changes in sediment fluxes could result in significant changes in barrier morphology. We also demonstrate that modeled barriers with reduced vertical sediment accommodation are highly sensitive to the magnitude and direction of shoreface fluxes. Therefore, natural barriers with limited sediment accommodation could allow for exploration of the future effects of sea-level rise and changing flux magnitudes over a period of years as opposed to the decades required for similar responses in sediment-rich barrier systems. Finally, because our model creates stratigraphy generated under different input parameters, we propose it could be used in combination with stratigraphic data to hindcast the sensitivity of existing barriers and infer changes in pre-historic morphology, which we anticipate will provide a baseline to assess the reliability of forward modeling predictions.



## 1 Introduction

Despite historically unprecedented rates of modern sea-level rise (SLR) and regional-scale coastal interventions affecting alongshore sediment redistribution, impacts to the morphology of barrier coasts over decadal to centennial scales remain loosely quantified. This information gap, resulting from an absence of both data and models with appropriate temporal resolution, contributes to a lack of long-term coastal management policy (McNamara and Lazarus, 2018), and in some cases could lead to either over- or under-estimation of decadal-scale coastal change. Especially in unmodified and semi-natural

barrier systems, the shoreface, beach, and backbarrier are likely coupled through cross- and along-shore sediment transport pathways over decadal timescales (Ashton and Lorenzo-Trueba, 2018), suggesting that simple models of sandy shoreline retreat that do not account for such interactions may significantly depart from real-world rates of transgression (Cooper et al., 2020).

        Historically, the modeling gap has evolved out of a tendency to view geomorphic change as a function of process-

scale and/or structural drivers. In the former, individual processes on the order of days to weeks become the dominant agents of morphologic change. These include high-water and storm events (Cohn et al., 2019), as well as bar welding events (Aagaard et al., 2004) and other processes that affect or are mediated by the surf and intertidal zones. In the latter, geologic controls over broad spatiotemporal domains (100s to 1000s of years / 10s of km), such as antecedent slope (Masetti et al., 2008; Murray and Moore, 2018; Raff et al., 2018; Shawler et al., 2020) and internal system dynamics play a larger role. Internal dynamics include,

for example, lags in the response of the shoreface to landward-directed overwash (Lorenzo-Trueba and Ashton, 2014; Ashton and Lorenzo-Trueba, 2018), which may also amplify or dampen the long-term retreat response of a barrier system to changes in SLR (Ciarletta et al., 2019a).

        At the decadal to centennial scale, underlying substrate sedimentology and shoreface sediment availability are among the most significant drivers of morphological change (Psuty, 2008; Brenner et al., 2015; Cooper et al., 2018). Similarly,

sediment budgets alter barrier landscapes on scales of meters to kilometers over years to decades, reflecting the sum of numerous events (*e.g.* storms, changes in wind/wave direction, etc.) occurring over sub-annual timeframes (Sherman, 1995). Subsequently, the construction of a mesoscale model to examine sediment availability requires a morphodynamic framework, or a model that utilizes sediment fluxes to drive changes in morphology. However, such a model need not be event-based to approximate the net result of flux-driven changes in time, allowing for reduced-complexity simulation (French et al., 2016).

Furthermore, by idealizing the geometry of the barrier system, it is possible to partition sediment volume within a simple deterministic framework, relying on geometric and algebraic relationships to shape the morphology of the system as a function of not just sediment fluxes, but changes in accommodation due to SLR. Such morphological relationships would be partly reliant on a set of geometric rules, in this case informed by conceptual models of subaerial barrier-island morphological change (Psuty, 2008; Psuty and Silveira, 2013).

Working toward understanding and modeling barrier behavior over decades to centuries, we use a reduced-complexity morphodynamic model to evaluate the cross-shore morphological and behavioral response of a barrier system given 1) variable



rates of sediment delivery/partitioning, 2) sea-level rise rate, and 3) sandy-substructure accommodation, or the vertical depth to which sediment can fill or erode to affect extension or retreat of the shoreline. The primary objective is to explore how combinations of these parameters beget different barrier states, including transgression, aggradation, and progradation. Ultimately, our goal is to understand how changes in sediment fluxes to/from the shoreface and dune, resulting from either SLR or variations in vertical accommodation, impact the magnitude and timing of barrier state transitions. We believe a record of the magnitude and directionality of changes in past sediment budgets could be reconstructed for real-world barriers with this and similar model frameworks to reveal the impact of such state transitions on modern and future barrier behavior. Combined with available historical data from modern barriers, which provide limited control points to verify morphological evolution (Cooper et al., 2018), our approach allows us to quantify, to the first order, the sediment flux and partitioning conditions driving a diverse spectrum of decadal and centennial changes in barrier morphology. We can also potentially capture transitions that may not have been observed or inferred in historical records.

## 2 Background

Whole-barrier dynamics at the mesoscale (10s to 100s yrs) are poorly represented by models, partly because the complexity of geomorphic processes at this scale cannot be easily represented by simple linear relationships, as well as because of a lack of historical and geological data (Cooper et al., 2018; Shawler et al., 2020). For example, historical records have been instrumental in most studies of real-world barriers, but these records are usually temporally incomplete or only focus on one moving boundary within the barrier system (*e.g.*, shoreline; Cooper et al., 2018). Despite this relatively poor resolution, the behavior and morphology of barriers as a function of sediment input at this scale have been qualitatively described by geomorphologists (Psuty, 2008; Psuty and Silveira, 2013). However, comparatively little work has been performed to quantify behavioral and morphologic thresholds from sediment input fluxes over decades to centuries, especially in combination with external controls (*e.g.* rate of sea-level rise) and geologic controls (*e.g.* sediment accommodation, substrate lithology; see Shawler et al., 2020).

Psuty (2008) outlines a conceptual model of sediment availability-driven morphological response in barriers where subaerial development is related to directionality and magnitude of shoreline migration (Figure 1). An increasingly positive sediment budget (sediment budget > accommodation creation) should lead to rapid progradation of the beach, and result in a succession of preserved foredunes formed in the wake of the receding shoreline. Coincidently, growth in net sediment budget should also lead to the formation of progressively diminutive foredune ridges, which become smaller due to the more limited time available for sediment accumulation in a single ridge. Conversely, an increasingly negative budget should result in more rapid transgression and a loss of foredune volume owing to erosion and a lack of time to accumulate new sediment as the shoreline migrates landward into relict ridges. The Psuty (2008) continuum additionally identifies an equilibrium state where the beach sediment budget approaches zero. When this occurs, the shoreline position stabilizes, allowing the gradual addition



of sediment to the active foredune. In such cases, the barrier can reach maximum subaerial development, potentially forming very large foredunes.

Motivated by the concepts of Psuty (2008), Ciarletta et al. (2019b) developed a two-step, cross-shore sediment partitioning model in which shoreface fluxes to the beach drive shoreline progradation, while foredune fluxes control the transport of sand from beach to dune. Within this framework, increasing accommodation from SLR removes volume from the subaerial system and forces erosion of the shoreline. Foredune volume is stored in an idealized, triangular profile, and the initiation of an incipient foredune ridge as the shoreline progrades is captured by a critical ridge spacing (Ciarletta et al.,

2019b). By modulating shoreface sediment fluxes, it is possible to affect changes in foredune size, approximating the response of foredunes as observed in nature at various sites around the world (e.g., Bristow and Pucillo, 2006; Nooren et al., 2017; Oliver et al., 2019). Moreover, by testing different combinations of shoreface and foredune flux magnitudes and comparing to real-world ridge heights, volumes, and shoreline position, it is possible to quantify past changes in the sediment budget (Ciarletta et al., 2019b).

Though a significant first step in reproducing decadal-centennial coastal behavior, the earlier model focused specifically on prograding barrier systems, ignoring not only transgression, but also the dynamics of the sandy backbarrier of barrier islands (Ciarletta et al., 2019b). It is well documented at places like Fire Island, New York that backbarrier strata beneath the modern island contain preserved beach, overwash, and marsh facies (Sirkin, 1972; Leatherman, 1985), indicating the barrier has both migrated and changed its width through time. Additionally, Raff et al. (2018) and Shawler et al. (2019)

document marsh migration onto the bayside and low-lying interior of formerly progradational barrier islands, indicating a relationship between barrier width and backbarrier drowning, such that increasing island width restricts the flow of sediment to the backbarrier, thereby increasing passive interior flooding from sea-level rise. This suggests that comparing the modeled backbarrier-marsh interface through time to stratigraphic information from real-world barriers could yield insights into the sediment flux conditions at the shoreline, even if past shoreline geometries are not fully recorded in the geological record.

The inclusion of transgression within an upgraded sediment partitioning model could allow for exploration of the full spectrum of barrier responses to changing sediment supply (Psuty, 2008) and perhaps barrier behaviors not well-described by previous conceptual models. For instance, despite the suggestion that rapid transgression ensures the cessation of dune maintenance (Psuty, 2008), sandy beaches and barriers exist that maintain both rapidly eroding shores and very large (>10 m height) foredunes. An active, extreme example of such a system includes part of the Indiana Dunes, along the southeastern

shore of Lake Michigan. The largest dune in this system, Mount Baldy, is 36 m tall, more than half a kilometer long, and 250 m wide (Kilibarda and Shillinglaw, 2015); by some measures it meets the definition of an aeolian megadune (Pye and Tsoar, 2009). Over 80 years, Mount Baldy migrated inland 135 m, while the corresponding lake shoreline eroded 98 m (Kilibarda and Shillinglaw, 2015). Additional instances of relict (but active in the last 2 kyrs) coastal megadunes occur along the United States' east coast, including the Savage Neck Dunes along the Chesapeake Bay side of the Virginia Eastern Shore (Davis et

al., 2020; Figure 2), and Jockey's Ridge and other large backbarrier dunes in the Outer Banks of North Carolina (Havholm et al., 2004). A combination of active and relict coastal megadunes and transgressive dunefields can also be found throughout



the world, sometimes in distinct sets of sand bodies and ridges occurring over wide areas, such as along the barriers of the Rio Grande do Sul and Santa Catarina coasts of Brazil (Dillenburg et al., 2004; Vintem et al., 2006; Hesp et al., 2007). A model framework that accounted for transgression could explore the flux balances required to maintain these natural features, which we hypothesize could result from beach-to-dune fluxes greatly exceeding sediment delivery to the beach.


Finally, a morphodynamic model able to represent progradational, transgressive, and transitional behavioral/morphologic states could be used to assess the sensitivity of barriers—including beach-/dune-ridge plains, spits, tombolos, and different types of embayed barriers, in addition to barrier islands—to changes in sediment availability under a range of different environmental and geologic forcing conditions. At the mesoscale, changes in vertical accommodation driven by the rate of sea-level rise and variations in antecedent geomorphology are suggested to play a major role in barrier response (Cooper et al., 2018; Shawler et al., 2020). Moreover, sandy-substructure accommodation (the vertical space needed to be filled or eroded to invoke shoreline migration over decadal scales) differs across the globe due to both local geology and available wave energy, suggesting the baseline sensitivity of barriers to sediment input/loss magnitudes varies considerably. For example, modern barrier islands along the U.S. Gulf and Atlantic coasts feature accommodation depths on the order of 4-6 m (*e.g.* Parramore and Cedar Islands in Virginia—Shawler et al., 2019 / Raff et al., 2018; Brigantine Island, New Jersey—Shawler et al., 2020; Fire Island, New York—Schubert, 2009 / Leatherman, 1985) to less than three 3 m (west-central Florida—Locker et al., 2001 / 2002a,b,c,d). Interestingly, the barrier islands in west-central Florida are considered among the most morphologically variable in the world and are prone to rapid change (Davis and Barnard, 2003). Adjusting accommodation in a morphodynamic model would allow us to test the influence of this factor on barrier response sensitivity and explore how coincident changes in sediment flux magnitudes drive transitions between progradation and transgression. Such an approach could yield insights concerning the pace at which sediment-limited barriers respond to changes in external forcing, including the rate of sea-level rise.




## 3 Methods

### 3.1 Numerical framework

This study utilizes an extended version of the 'ridge and swale' cross-shore barrier framework of Ciarletta et al. (2019b), which we refer to as the Subaerial Barrier Sediment Partitioning (SBSP) model. The SBSP model assumes the same underlying controls on morphology as implemented in the earlier framework: the partitioning of sand to the subaerial barrier is governed by the rate of sand delivery to the beach through cross and alongshore fluxes (*e.g.*, shoreface fluxes), as well as the flux of sand from the beach to the foredune (*e.g.*, foredune fluxes). However, instead of representing foredune volume with a discrete triangular profile, the SBSP model reduces the entire subaerial barrier into a landward-tapering profile (Figure 3; state variables and input parameters, see Tables 1 and 2). In this way, the model captures not only the foredune, but also the entire subaerial sandy backbarrier platform.





To simulate new foredune-ridge development, the barrier surface can also be segmented into multiple triangular profiles which are capable of cross-shore amalgamation (a combination of swale-infilling and lateral growth) to form composite foredune structures (Figure 4). As with the model of Ciarletta et al. (2019b), the formation of new foredune crests under prograding shoreline conditions occurs at a regular critical spacing $L_C$. It is also assumed that incipient foredunes cannot form directly on the shoreline (see Durán Vinent and Moore, 2013), requiring a characteristic setback distance $L_S$ (see Ciarletta et al., 2019b for additional discussion).

From the idealized geometry shown in the model setup (Figure 3), changes in the barrier system can be explained by just two state variables: the horizontal shoreline position $x_s$ and the profile volume of the active subaerial barrier A. These boundaries change in response to modulation of shoreface fluxes $Q_S$ and foredune fluxes $Q_D$, respectively.

Specifically, the shoreline position is given by

$$\frac{dx_s}{dt} = \frac{Q_S}{D_T} - \frac{Q_D}{D_T} - \frac{(x_s - x_f) \cdot \dot{z}}{D_T} \tag{1}$$

where $Q_S/D_T$ and $Q_D/D_T$ are the sediment delivery to the beach and to the foredune, respectively, divided by the time-dependent depth of accommodation at the shoreface $D_T$. The final term represents the loss of beach volume to sea-level driven accommodation creation ($\dot{z}$ is rate of sea level rise, and $x_s$ - $x_f$ is cross-shore width of the beach).

The cross-sectional volume $A$ of the subaerial barrier is a function of foredune fluxes $Q_D$ and the loss of subaerial volume to the subaqueous domain as sea level $Z$ rises. Specifically, the volume of the subaerial barrier is given by

$$\frac{dA}{dt} = Q_s - (x_b - x_f) \cdot \dot{z} \tag{2}$$

where $x_b$ - $x_f$ is width of the barrier subaerial profile, excluding the beach.

From $A$, and given the front and back slopes of the subaerial barrier ($\Gamma_1, \Gamma_2$), it is possible to calculate the height of the foredune crest as H = $[2 \cdot A/(1/\Gamma_1 + 1/\Gamma_2)]^{1/2}$. As H updates through time, the values of $x_b$ and $x_f$ can similarly be computed algebraically by

$$x_f = x_c - \frac{H}{\Gamma_1} \tag{3}$$

$$x_b = x_c - \frac{H}{\Gamma_2} \tag{4}$$

where $x_c$ is the active foredune crest location. The crest location is supplied as an initial input but updates as new foredune crests are created or old crests are reactivated through exposure to the active beach coincident with shoreline transgression. During foredune creation, a new $x_c$ is established once $(x_s - x_f)$ + H/$\Gamma_1 \geq L_C + L_S$, or when the combined width of the beach and seaward flank of the subaerial barrier profile are greater than the critical spacing plus the setback distance.

Unlike in Ciarletta et al. (2019b), all the equations can now run 'backwards', simulating erosion and transgression by accepting negative flux values until all volume has been exhausted from the barrier profile. To implement subaerial volume



loss through shoreline transgression, a simple geometric rule is applied. Erosion that undercuts the subaerial profile scarps the profile at a slope of $\Gamma_1$ (Figure 5). Volume scarped from the subaerial profile is then conserved to the beach, having the effect of counteracting negative shoreface fluxes. No volume is lost directly from the subaerial domain to outside the system—all volume is conserved unless removed by negative shoreface sediment fluxes.

We solve equations (1) to (4) using the Euler method over decadal to centennial scales and at an annual time step. Starting geometry is provided by the model input parameters outlined in Table 2 and described for individual investigations in the next section. The beach in our simulations is flat and maintains elevation with sea level $Z$. Where the width of the barrier is calculated in our results, we compute this term as $x_s - x_b$. This is different from the width of the subaerial profile coincident with $A$, which is calculated as $x_f - x_b$.

The initial input parameters used to describe the modeled barrier used in our investigations are shown in Table 3. In terms of the barrier superstructure, critical ridge spacing $L_C$ and front slope $\Gamma_1$ are informed by parameters observed at Fishing Point (southern spit end of Assateague Island) and Parramore Island (Ciarletta et al., 2019b), with the main difference being a slightly larger spacing to acknowledge some of the wider swales between relict foredunes visible at other U.S. East Coast barriers, including Fire Island, New York. In terms of the nucleation location of a new foredune ($L_s$), Durán Vinent and Moore (2013) note that, at least initially, the maximum height achievable by a foredune scales as a function of setback. Assuming a stationary barrier similar to the central regions of Fire Island and Assateague Island, a 3- to 5-meter tall aeolian dune could develop for a reasonable range of wind-induced shear values when $L_s$ is approximately 30 m. Subsequently, we also start the model with an initial beach width ($x_s - x_f$) equal to $L_s$. For the last parameter affecting the subaerial geometry, $\Gamma_2$, our modeled barrier has a backbarrier width ($x_f - x_b$) that scales an order of magnitude larger than the distance between the foredune crest $x_c$ and the effective base of the dune, $x_f$.

In the substructure, our model barrier uses initial $D_T$ values (sandy-substructure accommodation) in the range of 2.5 to 5 m. The smaller value was selected based on the sandy platform depths seen in west-central Florida barriers (Locker et al., 2001; 2002a,b,c,d). The larger value is the same as that used in Ciarletta et al. (2019b), and is based on stratigraphic data from Parramore Island and Fishing Point, Virginia (Halsey, 1978; Raff et al., 2018; Hein et al., 2019).

Our range of tested values for $Q_S$ and $Q_D$ is similarly inspired by real-world field sites. Himmelstoss et al. (2017) notes that long-term beach accretion along U.S. southeastern and Gulf coast barrier islands generally occurs on the order of 8.5 to 33.5 m yr$^{-1}$. If assuming a reasonable sandy substructure thickness (~5 m), this would be equivalent to shoreface fluxes on the order of 43 to 168 m$^3$ m$^{-1}$ yr$^{-1}$. Globally, beach-ridge plain systems tend to have slower rates of extension, on the order of 0.4 to 1.4 m yr$^{-1}$, scaling to 2 to 7 m$^3$ m$^{-1}$ yr$^{-1}$ (Bristow and Pucillo, 2006: Brooke et al., 2008; Hein et al., 2016). In the subaerial domain, a global compilation of field sites suggests foredune fluxes may occur on the order to 0 to 40 m$^3$ m$^{-1}$ yr$^{-1}$ (Ciarletta et al., 2019b, supplement), although we test somewhat higher values to account for cases where $Q_D$ may greatly exceed $Q_S$ by more than an order of magnitude.



## 3.2 Exploration of model behaviors

We explore the morphology and behavior of simulated barriers within the SBSP framework through two principle lines of investigation: 1) quantifying thresholds of behavioral and morphological change based on combinations of shoreface sediment fluxes $Q_S$ and variable rates of SLR ($\dot{z}$), and 2) exploring the full spectrum of barrier transgressive behavior, which is hypothesized to be richer than as conceptualized by Psuty (2008) (Figure 1) based on a number of field sites that show evidence of dune dominance, or net accumulation of sediment in the foredunes, despite shoreline erosion. The latter investigation tests

different combinations of $Q_S$ and $Q_D$, allowing $Q_D$ to exceed $Q_S$ by over an order of magnitude—the inverse of what was previously examined in Ciarletta et al. (2019b), which focused purely on scenarios of neutral to positive beach sediment budget. Within both lines of investigation, we also evaluate different sandy-substructure accommodation depths ($D_T$) to compare the sensitivity of sediment-rich and sediment-starved barriers to the same range of flux magnitudes.

In each investigation we explore impacts to barrier morphology by examining the state of four variables: the width of

the barrier $W$ (where $W = x_s - x_b$), the number of foredune crests $N$, the height of the active foredune crest $H$, and the location of the barrier-marsh interface $x_b$. A result of $N > 1$ allows us to quickly diagnose that progradation has occurred, and in combination with $W$ and $x_b$ can help illuminate the impact of backbarrier drowning. Where $N = 1$, we can identify stable and transgressive dunes, determine if they are growing or losing volume by examining $H$, and determine if inland migration has occurred by comparing $x_b$ and $W$. Note that $x_s$ is also known as long as $W$ and $x_b$ are tracked. To capture the loss of foredune

volume through time under transgressive regimes, we also initialize the model with a modest 2 m high dune.

## 3.3 Visual quasi-stratigraphic output

The model explorations outlined in the previous section are sensitivity analyses that rely on regime plots of $N$, $W$, $H$, and $x_b$ to interpret morphology and behavior. To demonstrate model behaviors in a two-dimensional cross-shore sense, we describe an example barrier evolution with graphical outputs which depict morphology as shown in Figures 3, 4, and 5. Here, we use the

model inputs for the $Q_S$ vs SLR investigation (Table 3) but hold $\dot{z}$ constant at 6 mm yr⁻¹ and set the initial dune height to 0 m while modulating $Q_S$ with a sine function to simulate an oscillatory sediment flux. We center our sine function on a flux of +7 m³ m⁻¹ yr⁻¹, with an amplitude of 50 m³ m⁻¹ yr⁻¹, such that the maximum flux is 57 m³ m⁻¹ yr⁻¹ and the minimum flux is -43 m³ m⁻¹ yr⁻¹. The period of the oscillations is set to 200 years (Figure 6), comparable to the magnitude of timescales over which significant changes in sediment flux are inferred at some U.S. east coast barriers (Leatherman, 1985; Deaton et al., 2017;

Ciarletta et al., 2019b).

Ten years after initialization (Figure 7a), our dune-less example barrier develops a significant subaerial superstructure with a height of 1.4 m. Height growth is rapid in the first few years due to a high ratio of accumulation surface to cross-sectional volume; as time progresses and the superstructure of the barrier enlarges, the rate of vertical growth slows. This mirrors the relationship observed by Davidson-Arnott et al. (2018) that, assuming a relatively constant sand source and flux,

dune growth slows (but never reaches zero) as the dimensions of the dune increase with time.



At 75 years into the model run, positive sediment fluxes result in progradation of the barrier and subsequent development of a new foredune ridge seaward of the original ridge. The sandy superstructure also built landward into the marsh prior to creation of the new foredune, burying old marsh surfaces previously adjacent to the backbarrier. Now that the original foredune has become relict, the flow of sediment to the backbarrier is cut off, and the marsh is beginning to passively

drown the backbarrier surface as sea level continues to rise (Figure 7b).

One hundred and five years into the simulation (Figure 7c), fluxes at the shoreface reach zero, but for the latter 30 of those years decreasing fluxes have been unable to compensate for increasing accommodation driven by sea-level rise at the front of the barrier, resulting in landward migration of the shoreline. The seaward foredune continued to grow over the same timeframe, encroaching and filling the swale separating it from the landward/relict foredune. In the current time step (105

years), the seaward foredune has filled all remaining swale accommodation and merged with the landward foredune, completing the process of cross-shore amalgamation. Now, as sediment fluxes become negative (triggering accelerated erosion and landward shoreline migration), the subaerial superstructure will become scarped.

At 200 years into the model run, now with exposure to 100 years of net negative shoreface fluxes, the barrier has undergone erosion on its seaward flank and passive drowning (marsh encroachment) on its landward flank (Figure 7d). Its

total width, which peaked at 837 meters 120 years earlier, is now only 384 meters. Evidence of earlier progradation at the front of the barrier has been eroded away, but the interfingering of sand and marsh facies in the backbarrier substructure indicates that the barrier dimensions were once larger. As shoreface fluxes become positive, the barrier will redevelop, although from a position 120 meters landward of where it was initialized. Despite having a slightly net positive sediment flux, transgression driven by sea-level rise has increased vertical accommodation, forcing net retreat overall.

At the end of the model run, 500 years later, the barrier has undergone three cycles of progradation and two cycles of transgression and drowning (Figure 7e). Evidence of prior constructive and destructive phases of morphologic evolution are buried beneath the contemporary barrier in the form of sand-marsh interfingers. However, on the surface, the modern barrier might only appear to be a long-term transgressive system that has recently undergone progradation. If the model run was extended for another 500 years, some of the stratigraphy preserved under the barrier would be eroded by the shoreface,

destroying the record of the earliest phases of progradation and transgression.

# 4 Results

## 4.1 Shoreface flux vs. sea-level rise

Our investigation of shoreface flux versus SLR reveals a complex pattern of barrier morphological response as depicted through tracking of $x_b$, $W$, $H$, and $N$ (Figure 8). At 500 years of barrier evolution, most outcomes with negative sediment fluxes

result in 'destructive' transgression, characterized ultimately by complete loss of the barrier superstructure. We describe this simply as 'dune loss', although it could also be the beginning of barrier island disintegration if the sandy substructure undergoes drowning or is depleted of sediment by continued erosion. Such outcomes would be delayed if the starting subaerial volume





were larger (relative to our foredune modeled with an initial crest height of 2 m). Where sediment fluxes are positive, the vast majority of $Q_S$ fluxes between 0 and 32 m³ m⁻¹ yr⁻¹ bracket a stable (subaerial constructional) transgressive form of barrier

development, as well as predominantly aggradational to mildly progradational forms ('pre-amalgamation') where SLR is less than 12 mm yr⁻¹ and $Q_S$ is greater than 17 m³ m⁻¹ yr⁻¹. The entire region between 0 and 32 m³ m⁻¹ yr⁻¹ is strongly modulated by sea-level rise, with increasing SLR thinning and reducing the height of the barrier, especially for low rates of positive shoreface flux (*e.g.* 5 m³ m⁻¹ yr⁻¹compared to 25 m³ m⁻¹ yr⁻¹). These behaviors largely correspond with the barrier geometry demonstrated in Figure 7d.

A significant break in behavior and morphology is developed at $Q_S = 32$ m³ m⁻¹ yr⁻¹, where increasing shoreface flux results in the ability of the beach sand budget to adequately compensate for $Q_D$ fluxes and loss of volume to dune aggradation. In this region, sea-level rise still exerts an impact on barrier morphology, splitting outcomes into two morphological endmembers. Specifically, at high rates of SLR, mostly exceeding 5 mm yr⁻¹, multi-ridge propagation is impossible, as increasing accommodation in the shoreface prevents the shoreline from rapidly prograding. With slower progradation, cross-

shore ridge amalgamation (similar to Figure 7c; see also Figure 4) dominates the response of the barrier, with the number of ridge crests $N$ alternating between 1 and 2 through time as each new seaward ridge is allowed time to grow before merging with the previous, relict ridge. At slower rates of SLR, accommodation at the shoreface is filled much faster than it is lost to vertical accommodation, and true progradation prevails, with the shoreline able to extend rapidly seaward—this triggers the production of a series of relict foredune ridges in the cross-shore ($N \geq 2$).

A simplified 'map' of all the aforementioned responses is shown in Figure 9, which also shows the change in morphological outcomes if initial $D_T$ is adjusted from 5 m to 2.5 m. At a timescale of 500 years, differences in combinations of $Q_S$ and SLR that result in destructive transgression are negligible, as loss of the barrier profile largely occurs within the first 200 years for both scenarios. However, at centennial scales the impact of decreased sandy-substructure accommodation is noticeable in the division between progradational and amalgamative behaviors. At $D_T = 2.5$ m, and for the range of input fluxes

tested here, progradation can occur at up to 20 mm yr⁻¹ of SLR, whereas at $D_T = 5$ m progradation only occurs up to 12 mm yr⁻¹ in the highest input flux case. Additionally, for the shallower substructure depth, a new behavioral regime becomes apparent when SLR is less than 5 mm yr⁻¹ and $Q_S$ is between 0 and 20 m³ m⁻¹ yr⁻¹. This regime, 'dune dominance' is characterized by runaway subaerial growth as the reduced accommodation at the beach allows for gradual shoreline regression even under relatively low shoreface flux conditions, similar to the example of the Florida barrier islands (see Sect. 2). This

permits subaerial accumulation (reflected in increasing $H$ and $x_b$; see Figure 10), which halts only when the seaward foredune toe ($x_f$) meets the shoreline. When this occurs, subsequent scarping temporarily recycles sand back into the beach before it is fed back to the subaerial superstructure in the next time step along with any new shoreface fluxes. Investigation in the next section shows that such a response only occurs when $Q_D > Q_S$, and it is not apparent in the response regime for $D_T = 5$ m because the enhanced platform accommodation consumes volume for vertical aggradation faster than it can be partitioned to

the subaerial portion of the system.



## 4.2 Shoreface flux vs. foredune flux

For our investigation of shoreface flux vs. foredune flux ($Q_S$ vs. $Q_D$), which focuses on previously unexplored transgressive retreat behaviors (see Ciarletta et al., 2019b), we use plots of $x_b$ and $H$ to differentiate behavioral modes (Figure 11), supplemented by quasi-stratigraphic cross-shore model outputs (Figure 12). For both $D_T = 2.5$ m and 5 m, the plots demonstrate

a complex behavioral regime in the aggradational to transgressive region ($Q_D \geq Q_S$), largely controlled by 'dune dominance,' in which the subaerial portion of the system consumes and stores the majority of the barrier-system volume. In the simplest terms, as the magnitude of $Q_D$ increases beyond the magnitude of $Q_S$, there is a gradient whereby the subaerial barrier initially becomes dune-dominated aggradational (accumulating sediment in the foredune but with a relatively stationary shoreline) and gradually transitions towards dune-dominated transgression (still accumulating sediment in the foredune, but undergoing

landward migration simultaneously). Transitions from dune-dominated aggradation to transgression occur over a smaller range of $Q_D$ values at $D_T = 2.5$ m (less shoreface accommodation) than for $D_T = 5$ m (more shoreface accommodation), as the volume of sediment needed to grow or erode the beach is smaller for shallower $D_T$. Overall, this makes the barrier more sensitive to changes in fluxes and promotes dune-dominated transgression (compare the extent of transgression regimes in $Q_D$ axis for $D_T = 5$ m and $D_T = 2.5$ m) as state shifts towards the transgressive endmember are imparted by relatively small differences in $Q_D$.

We describe the gradient in greater detail by examining scenarios starred in the plot of $x_b$ for $D_T = 2.5$ m (Figure 11). To begin, we examine a relatively high shoreface flux ($Q_S = 48$ m³ m⁻¹ yr⁻¹) counterbalanced by a similar magnitude, but slightly larger, foredune flux ($Q_D = 60$ m³ m⁻¹ yr⁻¹) (Figure 12a). Over the course of 500 years, the subaerial barrier becomes increasingly inflated, its growth paused only by episodes of occasional scarping as the seaward flank grows into the shoreline (marked by rapid fluctuations in beach width). When this occurs, scarping results in a transient return of sediment to the

beach/shoreface, volume ultimately recycled back to the subaerial barrier (with some loss to minor seaward progradation and vertical aggradation).

Maintaining the foredune flux but reducing shoreface flux to $Q_S = 30$ m³ m⁻¹ yr⁻¹, the barrier enters a transitional mode of transgressive behavior. Initially, the greater imbalance in flux partitioning results in erosion of the shoreline, with scarping failing to fully compensate for volume losses at the beach/shoreface. However, scarping allows for transient periods of beach

recovery, followed by episodes of subaerial volume accumulation. As time continues, accumulated subaerial volume becomes large enough that scarping returns greater quantities of sediment to the beach/shoreface, slowing and eventually reversing shoreline erosion to complete a transition to dune-dominated aggradation (Figure 12b). This transition is more pronounced when the magnitudes of both $Q_S$ and $Q_D$ are reduced ($Q_S = 5$ m³ m⁻¹ yr⁻¹ and $Q_D = 22$ m³ m⁻¹ yr⁻¹) resulting in a longer period of initial scarping and landward transgression of the foredune crest (Figure 12d).

When foredune fluxes exceed shoreface fluxes by approximately half an order of magnitude or more (we describe this as $Q_D \gg Q_S$), the behavior of the barrier becomes dominated by transgression. For positive shoreface flux values, this results in a mode of dune-dominated transgression, exemplified by the scenario of $Q_S = 23$ m³ m⁻¹ yr⁻¹ and $Q_D = 60$ m³ m⁻¹ yr⁻¹ (Figure 12c). The flux imbalance driving this behavior is so large that any transient shoreline extension resulting from





scarping is immediately reversed by rapid transfer of volume to the subaerial system. With the subaerial seaward flank unable
to extend seaward, the foredune undergoes high-frequency cycles of truncation on its seaward margin and volume
accumulation, resulting in net landward advance. Hypothetically, this behavior could eventually result in transition to
aggradation if the subaerial system becomes increasingly enlarged, resulting in scarping events sufficiently voluminous to
slow shoreline retreat. However, as subaerial growth slows with time owing to a decrease in the ratio of accumulation surface
to cross-sectional volume, the time horizon necessary to achieve relative 'stability' greatly exceeds the 500-year time horizon
of our modeling exercises.

## 5 Discussion

The results of our modeling exercises suggest that a two-step partitioning in sediment fluxes to barrier could explain many of
the morphologies and behaviors observed in nature. Of note, our framework appears to simulate some understudied
morphologies/behaviors, including dune-dominated transgression (arising when $Q_S < Q_D$) and cross-shore amalgamation.
Further, we find that our model captures the magnitude of mesoscale changes in moving boundaries (*e.g.*, shoreline) observed
and inferred for real-world barriers. We discuss the response of the barrier in terms of the predominant allogenic drivers in our
model (SLR and shoreface sediment fluxes), as well as highlight the implications of flux partitioning ($Q_S$ vs. $Q_D$), with
particular attention to dune dominance ($Q_S < Q_D$).

### 5.1 Sediment availability and relative sea-level rise

Our modeling results demonstrate that the response of barriers to changing rates of SLR and shoreface sediment fluxes ($Q_S$)
are non-linear. While increasing SLR (up to 20 mm yr$^{-1}$) generally results in a state shift towards more transgressive behavior,
state shifts occur extremely rapidly across a relatively limited range of specific shoreface sediment flux thresholds (~0 to 33
m$^3$ m$^{-1}$ yr$^{-1}$; Figure 9). The effect is that changes in barrier sediment availability may be more important than SLR in dictating
how behavior and morphology will change over the mesoscale. This has implications for studies of shoreline change that rely
solely on SLR and mathematical relationships to model shoreline migration over decadal to centennial scales. For instance,
Vousdoukas et al. (2020) projected future rates of global sandy shoreline retreat using a model based on the principles of the
Bruun rule, utilizing sea-level rise rates estimated from the Intergovernmental Panel on Climate Change's RCP4.5 and RCP8.5
scenarios. Subsequent probabilistic modeling of shoreline retreat showed that 13.6% to 15.2% of global sandy coasts would
retreat more than 100 m by the year 2050, with 35.7 to 49.5% of coasts severely eroded by the year 2100. Taking into account
modern beach widths, the authors posited that close to half of all sandy beaches could be "extinct" by the start of the 22$^{nd}$
century. However, as highlighted by Cooper et al. (2020), it is worth noting that this approach assumes a constant sediment
supply and a relative lack of cross- and along-shore sediment transport pathways. Significant departures from predicted
shoreline geometries could occur along shorelines where human coastal interventions, natural lags in alongshore sediment
transport, and variations in SLR (due to uneven and accelerating global sea-level rise) combine to alter sediment fluxes to the



beach. In addition to assuming a constant sediment supply, Vousdoukas et al. (2020) also note that their work does not examine the availability of horizontal and vertical accommodation for beaches to migrate. Both considerations impact our model results significantly, and the ability for a coupled beach-dune system to migrate vertically and horizontally is certainly beneficial for system longevity. For example, our modeled barrier persists for over a century at rates of sea-level rise more than double the present-day global sea-level rise rate of 3.3 mm yr$^{-1}$, including in cases of moderately negative sediment input (Figure 13).

This suggests natural beaches can keep up with accelerated sea-level rise over centennial timescales, but with the potential for substantial horizontal translation and/or coincident barrier narrowing.

      To achieve the magnitudes of shoreline retreat discussed by Vousdoukas et al. (2020), our model requires at least a modest shoreface sand deficit. However, our simulated barrier initializes with only a 2 m tall dune, and therefore does not capture the counterbalancing effect of large magnitudes of eroded dune sand feeding the beach that would occur where

preexisting dunes are more substantial. We expect that either a greater shoreface sand deficit or significant increases in the rate of SLR would be necessary to match previously modeled magnitudes of shoreline retreat (this study and Vousdoukas et al., 2020) in systems with larger subaerial volumes. Additionally, the modeled barrier has a width of 400 m, and 'beach extinction' as it exists in our model (coincident with dune loss and—in the case of barrier islands—potential disintegration, since our model allows the beach to migrate), is delayed in cases of wider barriers, which provide a platform for new subaerial volume

accumulation as well as more space to buffer erosion. This is exemplified by formerly and presently wide barrier islands such as Cedar and Parramore islands in Virginia, USA. Despite experiencing an acceleration in relative SLR of 3 to 4 mm yr$^{-1}$ over the last century (Boon and Mitchell, 2015), these islands have recently or historically sustained kilometer-scale landward shoreline migration over decadal to centennial timescales (McBride et al., 2015; Deaton et al., 2017; Shawler et al., 2019). Similar longer-term and sustained narrowing of previously wide barriers has also been inferred at the Bogue Banks, North

Carolina, a system of formerly progradational islands that began to undergo net shoreline erosion approximately 1 kya (Timmons et al., 2010). The combination of our modeling results and observations from natural systems therefore suggest that net sand surpluses over geological to historical timescales that serve to enhance system volume storage may render barriers more resistant to periods of sediment deficit or accelerated SLR, particularly over the mesoscale.

      However, the resistance afforded by historical sediment fluxes may not exist for built systems, in which human

development effectively removes volume and space that could otherwise be used to buffer the impacts of erosion and increasing accommodation from SLR. For example, in most developed stretches of the New Jersey coast, natural dunes no longer exist, replaced with relatively low, artificial dunes that provide little volume storage (Nordstrom and Arens, 1998). Human construction of dunes and other infrastructure can also block transport of sediment to foredunes and other parts of the subaerial barrier surface (Miselis et al., 2016; Rogers et al., 2015; Costas et al., 2006), preventing the accumulation of new volume in

the barrier interior that could later be available to buffer future shoreline erosion. Given that our results show that major barrier state changes occur over a relatively narrow range of shoreface sediment fluxes, we posit that development-related disruptions in sediment availability are likely to combine with enhanced SLR to promote rates of shoreline migration that exceed the variability of geological and historical records over decadal to centennial scales.



Discounting human development, Psuty and Silveira (2010) note that individual barriers experiencing changes in SLR
could also experience lags in alongshore response, complicating our cross-shore-focused approach and that of Vousdoukas et
al. (2020). Typically, downdrift beaches tend to be accretional and updrift beaches erosional, with an intervening point of
relative stability and equilibrium (Psuty and Silveira, 2010). Such a gradient is reflected in barriers such as Assateague and
Fire islands, in which downdrift spit ends function as sediment sinks for eroding updrift beaches (Figure 14). As sea level rises,
applying a landward forcing in sediment-starved reaches, the point of equilibrium should translate downdrift (Psuty, 2008;
Davidson-Arnott, 2005). This could result in updrift areas experiencing increasing erosion, while transient enhancement of
progradation occurs in distal downdrift reaches (Psuty and Silveira, 2010). The lag in response from updrift to downdrift may
trigger an island-wide transition to a rotational mode of barrier displacement, which is common in many barrier islands around
the world (FitzGerald et al., 1984). We anticipate that coupling our cross-shore model in the alongshore could capture this
downdrift-cascading behavior, allowing us to estimate how and when it will affect changes on modern barriers. In the interim,
however, the current cross-shore approach allows us to conservatively simulate the baseline behaviors that could arise in the
absence of alongshore-propagating variations in shoreface flux mediated by increasing rate of SLR.

## 5.2 Dune-dominated transgression

The investigation of the competition between shoreface ($Q_S$) and foredune ($Q_D$) fluxes expands the earlier work of Ciarletta et
al. (2019b), which mostly examined cases where $Q_S$ was moderately larger than $Q_D$. When $Q_S$ greatly exceeds $Q_D$, the
morphology of barriers is dominated by multiple, prograding, low-relief dune ridges. As the magnitude of $Q_S$ approaches $Q_D$,
the morphology of the barrier superstructure becomes increasingly dominated by a large, singular foredune, or as our new
results demonstrate, cross-shore amalgamation of dunes through time. When $Q_S < Q_D$, our modeling suggests the barrier is
subject to complex state shifts through different modes of dune-dominated aggradation and transgression, where barrier volume
is preferentially stored in the subaerial superstructure as opposed to the subaqueous substructure.

The southern Santa Catarina coast of Brazil presents qualitative examples of all $Q_S$ vs. $Q_D$ conditions presented in our
model (Rodrigues et al., 2020). Here, a progradational, mainland-attached beach- and foredune-ridge plain ('strandplain') is
backed by a succession of large, transgressive dune ridges/dune fields, in which the volume of subaerial sand (stored in the
form of relict/stabilized dune systems) corresponds not only with the past rate and direction of shoreline migration (function
of $Q_S$), but also the inferred magnitude of past wind-driven flux (a primary driver of $Q_D$). Specifically, Rodrigues et al. (2020)
describe periods of sustained beach progradation, believed to be coincident with wet periods of enhanced fluvial discharge to
the coast, as consuming sand volume that would otherwise be sent to dunes, much like in our the $Q_S > Q_D$ scenario in our
model. In contrast, conditions in which $Q_S < Q_D$ are believed to have occurred at Santa Catarina during the Little Ice Age,
recorded by an episode during which beach ridges transformed into aeolian foredunes/blowouts as the climate dried and
became windier. This episode is likely coincident with shoreline transgression (Guedes et al., 2011). Intriguingly, these
inferences about system behavior during the Little Ice Age suggest that, not only is $Q_S$ variable in relation to $Q_D$, but $Q_D$ may





be further enhanced during episodes of climate-mediated low $Q_S$, creating a feedback to drive the system towards dune dominance.

While changes in flux partitioning ($Q_S$ vs. $Q_D$) can be observed in cross-shore morphology at places like Santa Catarina, it is also possible to observe alongshore gradients in dune dominance associated with sediment availability within
individual modern barriers, with shifts from transgressive dune-dominated morphologies (updrift) to progradational beaches with multiple low-relief ridges (downdrift). To the first order, this suggests our model is capturing the details of a more complex morphological response spectrum than recognized by Psuty and Silveira (2010), but hinted at by others. Model results show that, at rates of SLR < 5 mm yr$^{-1}$ and given reasonable $Q_S$ values (-10 to 40 m$^3$ m$^{-1}$ yr$^{-1}$; Ciarletta et al., 2019b) six different morphological states are possible, including dune-dominated transgression. This level of morphological complexity is found,
for example, within the tombolo-barrier connecting the bedrock-cored islands of Miquelon and Langlade, off the southern coast of Newfoundland. Transgressive, parabolic dunes 15-20 m high ('Les Butteraux') characterize its northern end and, continuing southward, transition into a short stretch of large, linear dunes, and then into a set of relatively small progradational ridges (Billy et al., 2013; 2014). The morphological gradient mirrors the local alongshore transport regime (updrift erosional and downdrift depositional) observed on Fire Island, New York, but offers an alternate, dune-dominated example (Figure 14).
We note that our model-field comparison is complicated by sites where our model framework suggests the $Q_S < Q_D$ relationship could create large, transgressive dune morphologies, but no such dunes are present. This may be due to significant aeolian deflation, which, though not captured in the SBSP model, we view as another important flux shaping barrier landscapes. A modern example of such deflation-mediated morphology exists in the bedrock-grounded barriers of the Outer Hebrides, Scotland (Cooper et al., 2012). Sediment within this 'wind-dominated' (cf. Pile et al., 2019) barrier system is largely stored in
thin, landward-stretching sheets of windblown sand up to 2 km wide (locally referred to as *machair*). The lack of large dunes in this system is ascribed to a dearth of suitable trapping vegetation, as well as insufficient sediment delivery to offset landward deflation on transiently developed foredunes (Pile et al., 2019). This suggests the Outer Hebrides system could represent a further transgressive endmember evolving under a three-step partitioning model of fluxes, which includes not just fluxes from shoreface to beach ($Q_S$) and from beach to dune ($Q_D$), but also dune-to-machair fluxes (wind-driven deflation, $Q_W$). Such a
tripartite system is certain to have feedbacks on barrier response that exceed the scope of the current work, although it is tempting to speculate that such partitioning could allow barriers to respond more rapidly and less destructively to increasing SLR than our modeled system.

Finally, our model does not consider vegetation feedbacks, which previous modeling by Durán Vinent and Moore (2013) suggests may alter the shape of the dunes/dunefields formed by aeolian action. Specifically, Durán Vinent and Moore
(2013) note that stronger vegetative forcing results in more linear, stable dunes, while sparse vegetation creates transgressive dunefields, or in some cases, sand sheets (Kasse, 1997; Pile et al., 2019). Intriguingly, Mendes and Giannini (2015), studying the same section of the Santa Catarina coast as Rodrigues et al. (2020), posit that stabilization of dunes is tied to coincident increases in precipitation and decreases in windiness, which also suggests an inverse relationship between vegetation growth and $Q_D$ in natural systems. This relationship may enhance the morphological impacts of the inverse relationship between $Q_S$



and $Q_D$ implied by Rodrigues et al. (2020), suggesting that, at least on centennial scales, climate (via impacts to vegetation/land cover) is a major driver of barrier/beach behavior. Such a hypothesis is supported by the work of Jackson et al. (2019), who found that synchronous dune transgression and coastal erosion occurred in Europe during the Little Ice Age as a function of combined climate-mediated vegetation dieback, increasing windiness (due to increased storminess), and enhanced aeolian flux. Multiple systems of large, transgressive dunes in the USA, such as the Savage Neck Dunes in Virginia (Davis et al., 2020), were similarly active during the Little Ice Age (Havholm et al., 2004), potentially signifying a global expansion of dune-dominated coastal systems into the mid-latitudes during that time. In the context of modern climate change, this relationship between vegetation and $Q_S$-$Q_D$ partitioning suggests relatively high-latitude present-day barriers (*e.g.* those at Miquelon-Langlade) could be subjected to decreasing dune dominance in the future, since the formative climatic conditions that led to dune accretion may no longer exist. We posit that, as subaerial morphology becomes relict, erosion of the barrier superstructure could result in an irreversible state transition that gradually enhances system vulnerability to increasing SLR and decreasing sediment availability. In future study, this hypothesis could be investigated in an expanded framework that considers barrier sensitivity to changes in $Q_W$. Specifically, Jackson et al. (2019) describes a model of coastal dune behavior effectively driven by $Q_S$ and $Q_W$ that could be combined with our numerical implementation of the Psuty (2008) conceptual model ($Q_S$ and $Q_D$) to construct a quantitative multi-flux framework ($Q_S$, $Q_D$, and $Q_W$).

**5.3 Sandy-substructure accommodation**

Our model framework allows us to vary the depth of the barrier substructure, or its sandy platform thickness, to explore how it is affected by both sediment fluxes and an increasing rate of SLR. Specifically, we showed that decreasing the substructure depth from 5 m to 2.5 m increases barrier sensitivity to morphologic state shifts, as decreased vertical accommodation increases the corresponding horizontal shoreline response (see Figures 9 and 11). This relationship also enables some unexpected behaviors, especially as it relates to dune-dominated transgression. Under conditions of limited substructure depth, material eroded from dunes rapidly replenishes the beach, allowing beach-to-dune fluxes to persist even as the shoreline undergoes overall retreat (*e.g.* Figure 11c). The enhanced sensitivity to state shifts can be observed in the number of input parameter combinations ($Q_s$ and SLR) that result in amalgamative behaviors. Relative to $D_T$ = 5 m, the shallower $D_T$ = 2.5 m expands the number of input combinations where strictly progradation or aggradation/transgression are possible, reducing the extent of the amalgamative 'buffer' between the two end-member states. We also note that, with decreased $D_T$, the destructive transgressive regime is slightly expanded, particularly at slower rates of SLR (< 5 mm yr$^{-1}$). This occurs because even slightly negative fluxes result in greater horizontal (*e.g.*, landward) shoreline displacement than would occur with a deeper substructure.

Based on the results of our investigations, which do not account for cross-shore energetics, a possible consequence of shallower substructure depth is that alongshore sediment delivery could become the dominant mechanism influencing barrier states. In real-world barriers, there is likely a degree of similarity in the concept of a sandy substructure depth and the inner depth of closure, which is defined as the seaward limit of the 'littoral zone' (Hallermeier, 1981). While the latter is a cross-shore construct based on wave energy, Hallermeier (1978) describes the littoral zone as being significantly shaped by



alongshore transport, a process which at least one study has shown can cause upper-shoreface profiles to depart significantly from cross-shore equilibrium predicted by the Bruun rule (List et al., 1997). Similarly, historical and geological investigations
indicate that barriers subjected to kilometer-scale shoreline movement over multi-decadal timescales are strongly influenced by spatiotemporal variations in alongshore sediment delivery (Brooke et al., 2008; Lindhorst et al., 2010; Oliver et al., 2017; Raff et al., 2018; Shawler et al., 2019)

An example of this influence is provided by the barrier islands along the west-central Florida coast, which are described as being not only extremely dynamic, but also alongshore-transport dominated (Davis et al., 2003; Davis and
Barnard, 2003). With vertical accommodation limited by bedrock and scarce offshore sediment reservoirs (limiting the contribution of cross-shore transport), variations in alongshore sediment redistribution can lead to pronounced changes in coastal geography and morphological state over sub-centennial scales. Combining our model results with observations from places likes west-central Florida, we propose that barrier systems with limited vertical accommodation could make ideal laboratories to study the morphological impacts of changes in sediment input and SLR over relatively short timescales,
potentially predicting the magnitude of future changes in systems with slower responses. That said, we recognize that direct comparison between sites with different $D_T$ could be hindered by varying ratios of shoreface and dune fluxes or more complex regimes, in which aeolian deflation represents a significant loss. Consequently, a more accurate comparison might be drawn between barriers with different $D_T$ that exist at similar latitudes, because, as the work of Rodrigues et al. (2020) shows, subaerial fluxes are influenced by regional-scale climate variability.

## 6 Conclusions and implications

Our model investigations, coupled with historical and modern observations of real-world barriers, highlight the importance of sediment delivery and partitioning within coastal systems in driving their mesoscale behavior. Over the scale of decades to centuries, changes in direction and magnitude of sediment transport can cause the seaward and landward limits of barriers— as well as their subaerial height and topographic complexity—to vary on the order of meters to kilometers. Among other
outcomes, variations in sediment delivery and partitioning are expected to cause significant departures from shoreface equilibrium retreat geometries, suggesting that mathematical relationships defining rates of shoreline retreat as a function of rate of sea-level rise (and underlying shelf slope) are extremely restrictive at this timescale—especially if attempting to predict changes in subaerial morphology and resulting ecology.

Specifically, with our two-step partitioning approach, we demonstrate and infer the following:

1. Modeled barriers respond non-linearly to changes in RLSR (up to 20 mm yr[-1]), although variations in sediment availability play a much more significant role in driving morphological shifts over the mesoscale. Our framework also captures several behaviors that are not readily described by the classic progradation, aggradation, and transgression states, including amalgamation and dune-dominated aggradation/transgression.



2. Dune dominance in our modeled barrier can lead to a complex array of behaviors. Over decadal to centennial scales, dune-dominated barriers can be either aggradational or transgressive, or may exist in a notable transitional state. During dune-dominated transgression, the subaerial volume of the barrier can be continually inflated, despite landward retreat of the shoreline. This process is driven by a recycling of sediment scarped from the dune back into the beach, where it can be subsequently repartitioned back to the subaerial portion of the system.

3. Decreased vertical accommodation to extend or erode the sandy substructure of the barrier increases the sensitivity of the system to both increasing SLR and changes in sediment delivery and partitioning. This enhanced sensitivity manifests in both progradational and transgressive responses, including under conditions of dune dominance. Because barriers with limited vertical accommodation respond rapidly to external forcing, we propose that they could offer ideal natural laboratories to study the effects of changing SLR or sediment delivery over sub-decadal and sub-

centennial timescales.

Importantly, our work does not fully consider the impacts of aeolian transport, event-scale overwash, and vegetation, any of which could enhance or decrease the resistance of the barrier structure to changes in sediment delivery, as well as alter the partitioning of sediment in the cross-shore (*e.g.* plants increasing subaerial sediment trapping efficiency, or overwash

involving both landward- and seaward-directed sediment transport components). This effort does, however, suggest modeling morphologic change based on time-variable sediment input could be key to resolving the baseline state of barrier systems, especially those impacted by anthropogenic sediment interventions (*e.g.* nourishment, groins). Alternatively, from an ecological perspective, this could help inform management by refining understanding of a system's natural resistance to change, or the range of sediment flux variation and partitioning that can be accommodated without crossing a state threshold.

Although we do not account for all potential flux partitioning, our model results agree with the magnitude and timescale of changes in barrier morphology observed along the U.S. East Coast, suggesting that meter to kilometer-scale migration of horizontal cross-shore boundaries within barriers occurs readily as a function of mesoscale variation in sediment availability. These changes are sufficiently large to obscure the signals of long-term retreat trends over decadal to centennial timescales, and they illustrate the challenge of applying geometric relationships to predict, for instance, shoreline movement

as a function of increasing sea level alone. However, by examining relict geomorphology and historical changes in shoreline geometry, it may be possible to infer the magnitude of past changes in sediment delivery, as well as estimate differences in the flux partitioning between the subaqueous and subaerial barrier components. Coupling this approach with cross- and along-shore geomorphological observations from field and remote sensing efforts may ultimately allow quantitative exploration of barrier- and regional-scale sediment flux gradients over decadal to centennial timescales. Additionally, we hypothesize that

this approach could prove valuable to interpreting millennial-scale records of sediment delivery and accommodation changes preserved in the stratigraphy of long-lived beach-ridge plain systems, where our model could be applied to shorter (sub-millennial) time intervals to reconstruct system evolution in a piecewise manner.

## 7 Code and data availability

Example model data outputs generated during this study, as well as the model code used in these investigations, are available as part of an accompanying USGS software release (https://code.usgs.gov/spcmsc/subaerial-barrier-sediment-partitioning-model; Ciarletta et al., 2020).

## 8 Author contributions

D.C. conceived the model experiments, planned the investigations with J.M., carried out the simulations, and took the lead in writing the manuscript, with critical feedback from J.S. and C.H.

## 9 Competing interests

The authors declare no competing interests. Any use of trade, firm, or product names is for descriptive purposes only and does not imply endorsement by the U.S. Government.

## 10 Financial support

This work made possible through the USGS Mendenhall Research Fellowship program and the USGS Coastal and Marine
Geology program. A Virginia Sea Grant (NOAA) Fellowship (Agency Award NA18OAR4170083) supported J. Shawler. This paper is contribution #XXX of the Virginia Institute of Marine Science, William & Mary.

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

**Table 1 State variables for Subaerial Barrier Sediment Partitioning (SBSP) model**

| Symbol | Units | Description |
| --- | --- | --- |
| $t$ | T | Time |
| $x_s$ | L | Shoreline location |
| $x_f$ | L | Foredune toe location |
| $x_c$ | L | Foredune crest location |
| $x_b$ | L | Backbarrier-marsh interface |
| A | $L^3 L^{-1}$ | Active subaerial cross-section volume |
| Z | L | Sea level |






**Table 2 Input parameters for Subaerial Barrier Sediment Partitioning (SBSP) model**

| Symbol | Units | Description |
|---|---|---|
| $Q_S$ | $L^3 L^{-1} T^{-1}$ | Shoreface flux |
| $Q_D$ | $L^3 L^{-1} T^{-1}$ | Foredune flux |
| $L_C$ | $L$ | Critical ridge spacing |
| $L_S$ | $L$ | New foredune shoreline setback |
| $\Gamma_1, \Gamma_2$ | $L L^{-1}$ | Front and back slope of subaerial surface |
| $D_T$ | $L$ | Platform depth/depth to transgressive surface |
| $\dot{z}$ | $L T^{-1}$ | Rate of sea-level rise (SLR) |

**Table 3 Inputs investigated for Subaerial Barrier Sediment Partitioning (SBSP) model**

| Symbol | Qs vs. SLR | Qs vs. $Q_D$ |
|---|---|---|
| $Q_S$ | -20 to 50 $m^3 m^{-1} yr^{-1}$ | -10 to 50 $m^3 m^{-1} yr^{-1}$ |
| $Q_D$ | 30 $m^3 m^{-1} yr^{-1}$ | 0 to 75 $m^3 m^{-1} yr^{-1}$ |
| $L_C$ | 130 m | 130 m |
| $L_S$ | 30 m | 30 m |
| $\Gamma_1, \Gamma_2$ | 0.06, 0.006 | 0.06, 0.006 |
| $D_T$ | 2.5 to 5 m | 2.5 to 5 m |
| $\dot{z}$ | 0 to 2 mm $yr^{-1}$ | 1 mm $yr^{-1}$ |
| $H_{initial}$ | 2 m | 2 m |
| $t_{max}$ | 43 to 500 yrs | 500 yrs |

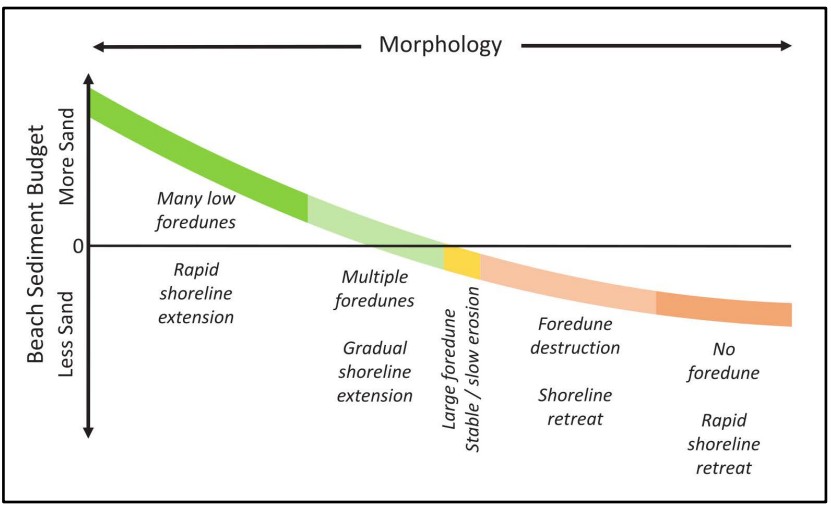

**Figure 1: Conceptual morphological continuum, depicting the subaerial characteristics of a barrier in response change in beach**
**sediment budget. Modified after Psuty (2008). Used with permission from Springer Nature.**





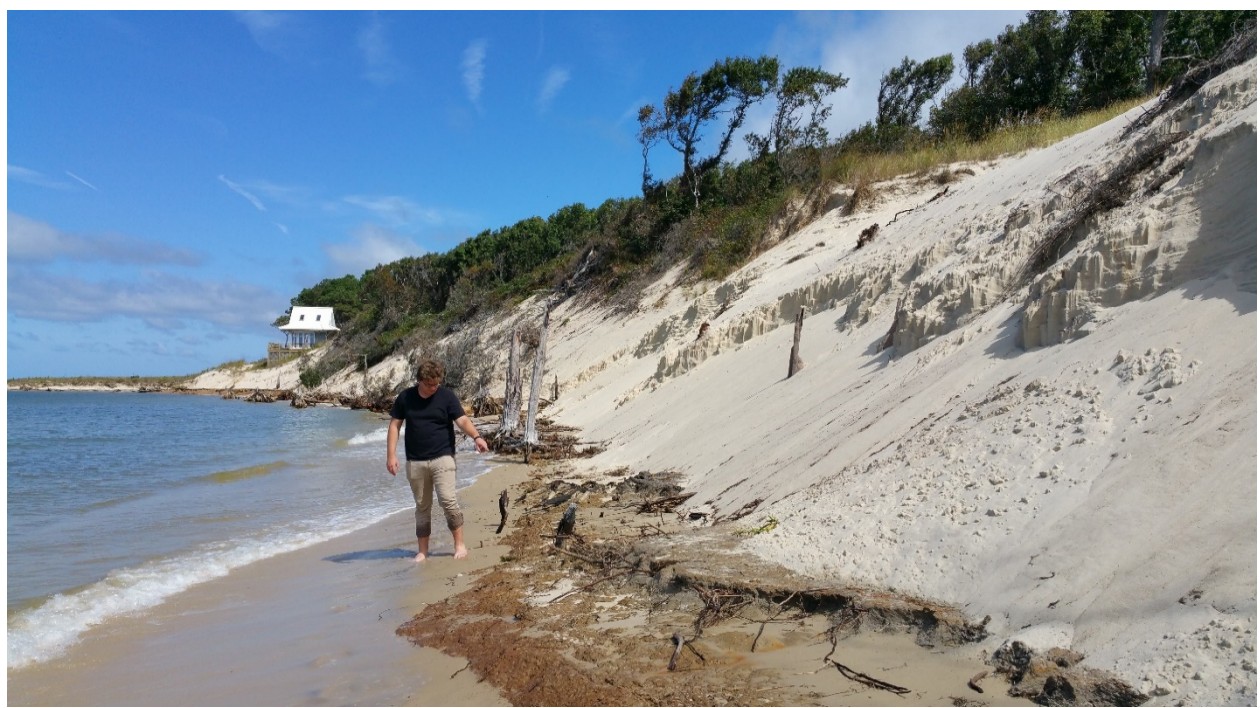

**Figure 2: J. Shawler inspects an erosional scarp in the Savage Neck Dunes, near Cape Charles, Virginia. Modern shoreline transgression has exposed portions of the basal terrestrial surface over which the dune system migrated, revealing the remnants of a late Holocene coastal forest buried by windblown sand. Radiocarbon dating of woody debris in the basal surface suggests dune**
**building persisted until several hundred years ago (Davis et al., 2020). Image by D. Ciarletta.**

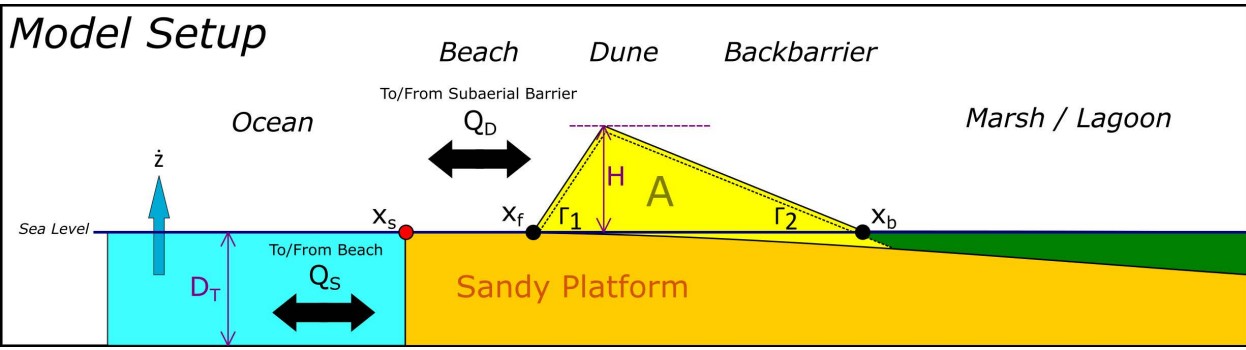

**Figure 3: Subaerial Barrier Sediment Partitioning (SBSP) model setup, depicting moving boundaries and processes. Black arrows indicate the bidirectionality of shoreface-to-beach ($Q_S$) and beach-to-dune ($Q_D$) fluxes, where positive $Q_S$ fluxes shift the shoreline (red dot, $x_s$) seaward and positive $Q_D$ fluxes increase the profile area ($A$) of the active subaerial surface (yellow-shaded region above**
**sea level), expanding the foredune toe ($x_f$) seaward, and the backbarrier-marsh interface ($x_b$) landward (black dots). Increases in subaerial cross-sectional volume increase the foredune crest height as function of subaerial geometry, governed by the front slope of the subaerial surface ($\Gamma_1$) and the back slope ($\Gamma_2$). Coincident with flux contributions/losses, subaerial cross-sectional volume is lost to the subaqueous domain (yellow-shaded region below sea level, not included in $A$), becoming part of the barrier substructure due to sea-level rise ($\dot{z}$). The thickness of the barrier substructure is denoted by $D_T$, the depth of the active shoreface over decadal**
**timescales.**





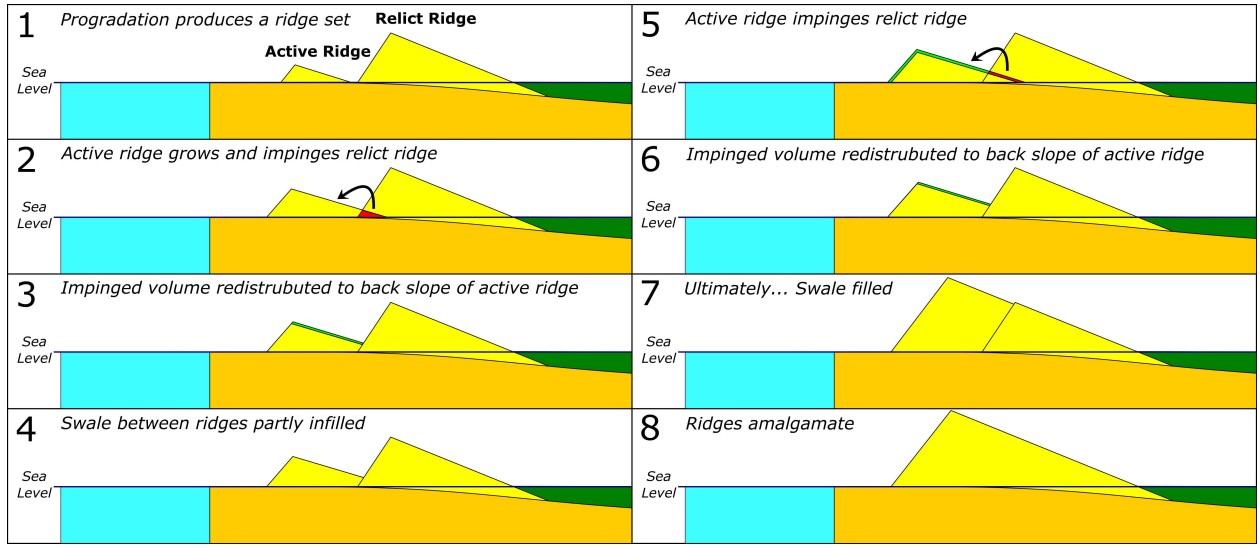

**Figure 4: Conceptualization of swale-infilling and cross-shore foredune ridge amalgamation in the Subaerial Barrier Sediment Partitioning (SBSP) model. As an active ridge grows and impinges a landward/relict ridge, the inflating ridge volume is redistributed to fill the swale between ridges. When swale filling is complete, the active and relict ridges amalgamate into a single active ridge. Yellow = barrier sand, blue = ocean, dark green = marsh.**

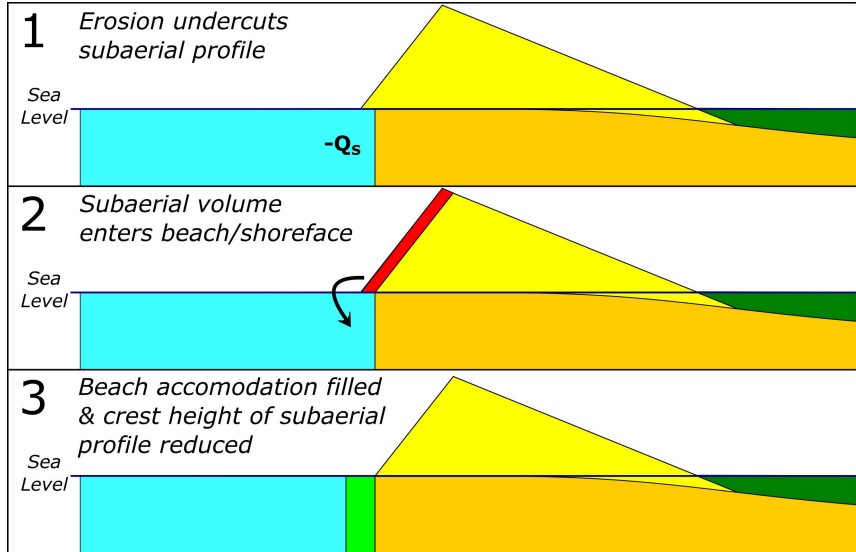

**Figure 5: Conceptualization of erosion and redistribution of volume in subaerial profile. As the shoreline erodes landward the subaerial profile is undercut and scarped according to the subaerial front slope $\Gamma_1$. Scarped volume is transferred to the beach/shoreface and fills accommodation to extend the shoreline seaward as a function of volume divided by depth. Some of this sand volume may then be lost from the system through continued beach erosion. Yellow = barrier sand, blue = ocean, dark green = marsh. $Q_S$ = shoreface flux.**



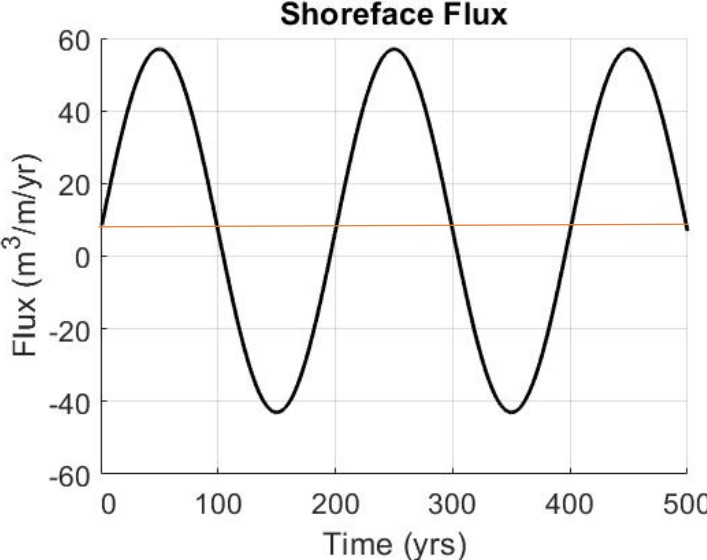

**Figure 6: Shoreface flux $Q_S$ supplied as a function of sine wave centered on +7 m³ m⁻¹ yr⁻¹ (net positive).**





**870** **Figure 7: Example simulation of a barrier subjected to a periodically oscillating shoreface sediment flux. Graphical output displays barrier at progressive timesteps of 10, 75, 105, 200, and 500 years post-initialization, undergoing state shifts between progradation and transgression. In the profile morphologies (left), yellow = barrier sand, blue = ocean, green = marsh. Solid magenta line tracks position of barrier-marsh interface, while dashed red line tracks the shoreline position.**





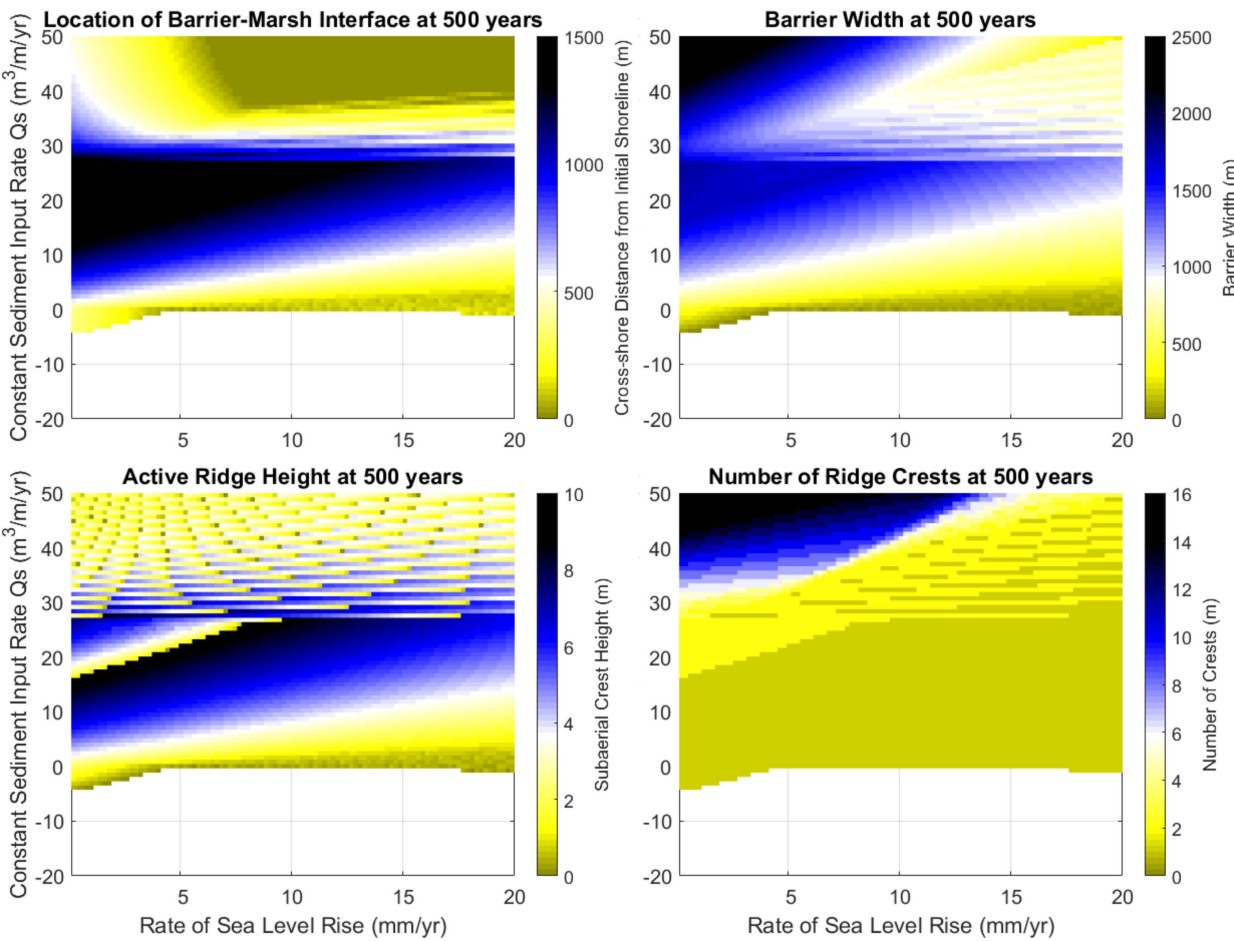

**Figure 8: Plots of barrier-marsh interface location $x_b$, barrier width $W$, active foredune crest height $H$, and number of foredune ridges $N$ for a 500-year simulation of a modeled barrier with input parameters described by Table 3 and $D_T = 5$ m. Regions with no data (generally negative sediment input) correspond with complete dune loss and potential barrier disintegration. $D_T$ is the thickness of the barrier substructure, or the depth of the active shoreface over decadal timescales. $Q_S$ = shoreface flux.**



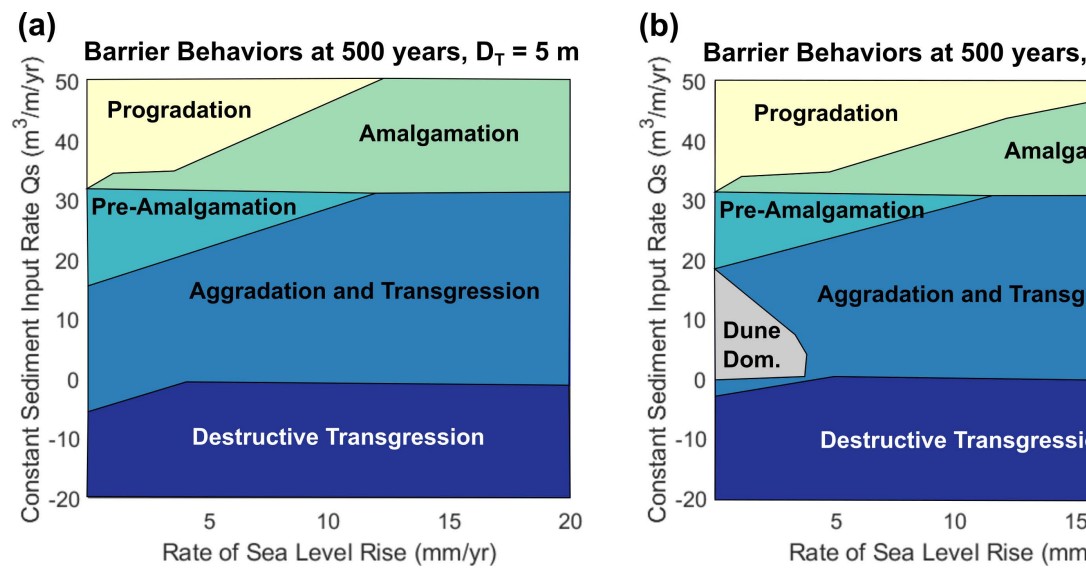


**Figure 9: Behavior of modeled barrier in response to combinations of $Q_S$ and SLR over 500 years for A) $D_T$ = 5 m and B) $D_T$ = 2.5 m. A region of $D_T$ = 2.5 m where SLR is less than 5 mm yr$^{-1}$ and $Q_S$ is between 0 and 20 m$^3$ m$^{-1}$ yr$^{-1}$ is subject to 'dune dominance', where the barrier volume is preferentially stored in the subaerial domain. $D_T$ is the thickness of the barrier substructure, or the depth of the active shoreface over decadal timescales. $Qs$ = shoreface flux.**


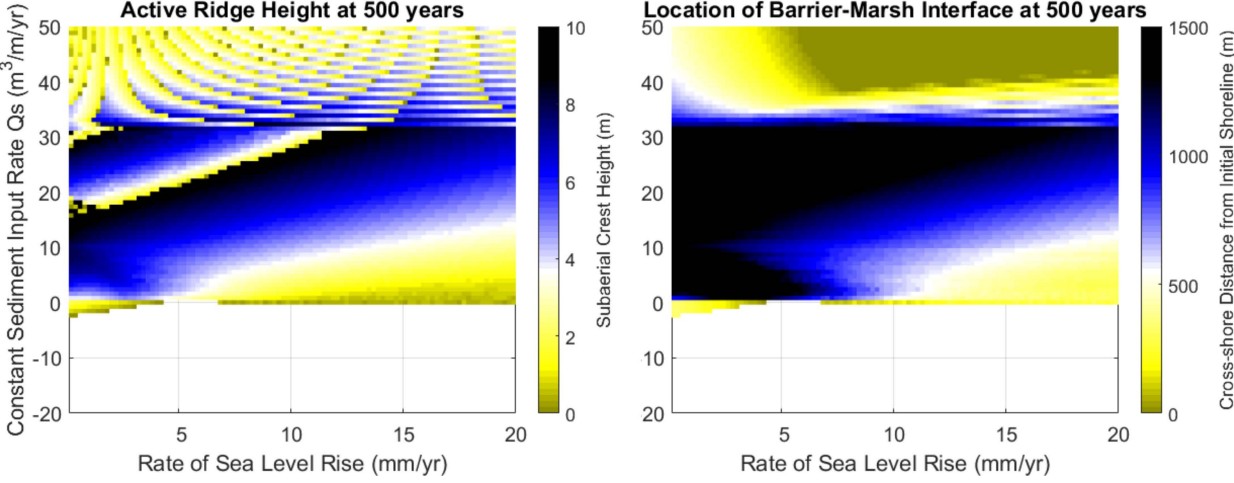

**Figure 10: Plots of active foredune crest height $H$ and barrier-marsh interface location $x_b$ for a 500-year simulation of a modeled barrier with input parameters described by Table 3 and $D_T$ = 2.5 m. Regions with no data (generally negative sediment input) correspond with complete destruction of the barrier's subaerial superstructure. $D_T$ is the thickness of the barrier substructure, or**
**the depth of the active shoreface over decadal timescales. $Q_S$ = shoreface flux and $x_b$ = cross-shore location of backbarrier-marsh interface.**





**Figure 11: Plots of barrier-marsh interface location *x_b*, and active foredune crest height *H* for a 500-year simulation of a modeled barrier with input parameters described by Table 3 and *D_T* = 5 m and 2.5 m. Regions with no data (destructive transgression) correspond with complete dune loss and potential barrier disintegration. D_T is the thickness of the barrier substructure, or the depth of the active shoreface over decadal timescales. *Q_S* = shoreface flux.**




**Figure 12: Simulated barriers at 500 years post-initialization. (a) Dune-dominated aggradation concomitant with 48 m³ m⁻¹ yr⁻¹ of shoreface flux and 60 m³ m⁻¹ yr⁻¹ of foredune flux. (b) Delayed dune-dominated aggradation at 30 m³ m⁻¹ yr⁻¹of shoreface flux and 60 m³ m⁻¹ yr⁻¹ of foredune flux. (c) Dune-dominated transgression at 23 m³ m⁻¹ yr⁻¹ of shoreface flux and 60 m³ m⁻¹ yr⁻¹ of foredune flux. (d) Delayed dune-dominated aggradation (pronounced initial transgression) at 5 m³ m⁻¹ yr⁻¹ of shoreface flux and m³ m⁻¹ yr⁻¹ of foredune flux. Yellow = barrier sand, blue = ocean, green = marsh. Solid magenta line tracks position of barrier-marsh interface, while dashed red line tracks shoreline position through time.**





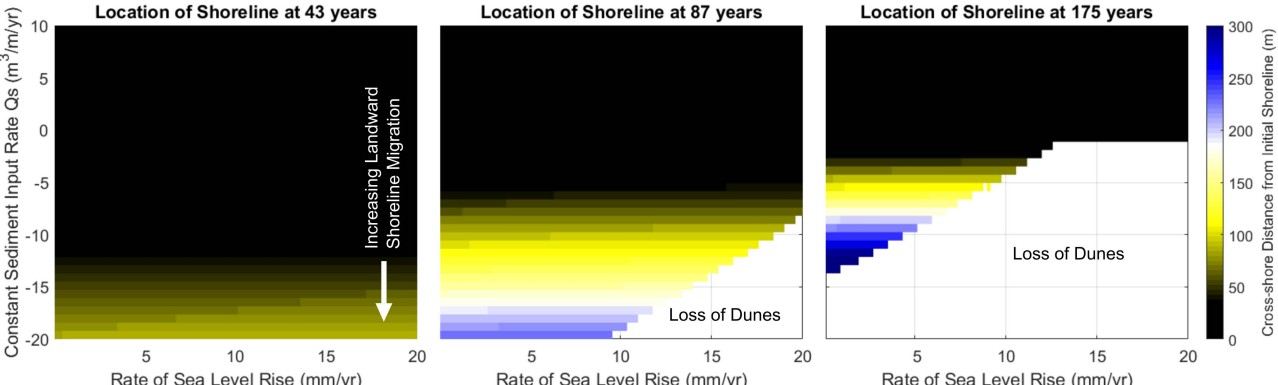

**Figure 13: Plots of shoreline location $x_s$ for a simulated barrier experiencing $Q_S$ in the range of -20 to 10 m³ m⁻¹ yr⁻¹ over time steps of 32, 87, and 175 years. Regions with no data correspond with complete destruction of the barrier's subaerial superstructure (loss of dunes). Note that deepest black shade represents a distance to the initial shoreline of ≤ 0 m. $Q_S$ = shoreface flux.**

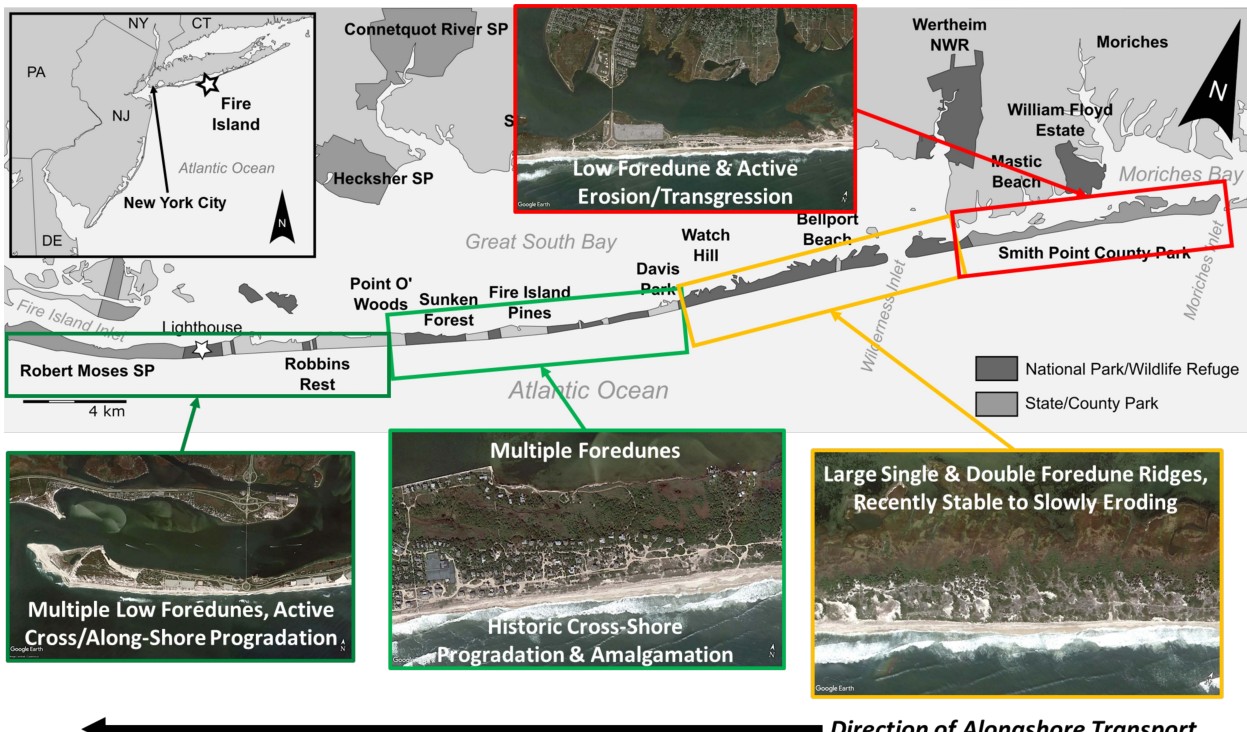


**Figure 14: Overview of Fire Island alongshore sediment transport gradient, with updrift (eastern) end of island displaying low, overwashed transgressive morphology and downdrift (western) end of island showing multiple beach ridges and spit extension typical of distal accretion. Inset shows location of Fire Island within the coastal Mid-Atlantic region. Green outlines correspond to regions characterized by present and/or historical progradation and cross-shore amalgamation, yellow to those characterized by aggradation or slow erosion, and red by erosion/migration. Map data © Google Earth, Landsat/Copernicus 2019.**
