# Peer review of "Quantifying Thresholds of Barrier Geomorphic Change in a Cross-Shore Sediment Partitioning Model"

_Earth Surface Dynamics, 2020_

## Referee Comment (RC1) · Eli D Lazarus (Referee) · 17 Dec 2020

REVIEW – esurf-2020-88 (Ciarletta et al.)

This work by Ciarletta & colleagues is an engaging and well-written exploration of cross-shore barrier dynamics in a deliberately simplified numerical model. The authors are clear about what the model does and does not explicitly address, and that it serves a tool for first-order quantitative insight into transitions between barrier states and behaviors otherwise framed in conceptual terms. The differentiated regime space indicates the rich dynamics that the model is capable of simulating, and the authors are careful to keep one foot in the real world by constraining the fundamental parameters

of the model with empirical rates.

My comments are minor, but I hope offer some helpful suggestions:

Abstract, Introduction – To me, snippets of the Abstract (and related snippets of the Introduction) are written in a way that suggests a decade is approximately the upper end of the model-world time scale here. (Other parts of the manuscript are clearer in this regard.) Just to reflect: the model runs on an annual increment and can comfortably tick over for a couple of centuries, which means the interesting changes occur on decadal time scales. If the authors describe it as a model of multi-decadal barrier dynamics – on the order of 101–102 years – I think that might help readers have a better sense from the outset of what the model does and doesn't do. (And it creates useful space for the interesting discussion, late in the manuscript, regarding what explicitly including event-driven changes and fast-acting processes might add.)

P3L66 and elsewhere – Recommend cutting "we believe," since it's implicit in the conditional statement that follows. Would also change instances of "believe" on P14L440/442 – prefer "thought to be. . ." or similar.

P3L76 – "Whole-barrier dynamics at the mesoscale (10s to 100s yrs) are poorly represented by models, partly because the complexity of geomorphic processes at this scale cannot be easily represented by simple linear relationships. . ." – I think Werner (2003) on complex hierarchy offers a helpful explanation here (because what manifests at the mesoscale is a mix of linear and nonlinear), and I'd offer that McNamara & Werner (2008a, 2008b) are one example of a barrier model that does quite well in this regard (and ultimately manages to account for both natural and human-dominated conditions). The operational blocks of this model (i.e., triangles, partitioned distribution) in many ways appear related, at least conceptually, to that earlier work. And even if the similarity ends there, it's a sound foundation. Furthermore, the spatially extended version of the LTA14 model exhibits some interesting mesoscale behaviour – which the authors cite (Ashton & Lorenzo-Trueba, 2018). All that is to say – I suggest the authors

cut back the "poorly understood" spin throughout the Introduction and take a lighter, more open tone like the one at P3L80 – a framing of the problem that makes plenty of room for previous work. Another reason for that subtle adjustment is that this work, too, is, as the authors rightly state, a "first-order" model, steered but hardly constrained by empirical parameters.

Sticking with this same introductory sentence – I'm not sure that geomorphic processes operate at the whole-barrier scale (and I think the authors would concur). Returning to Werner (2003) – whole-barrier dynamics are an emergent property of geomorphic processes of sediment transport at smaller-than-whole-barrier scales. So perhaps the authors could revisit their warrant here. My sense is that models struggle to represent whole-barrier dynamics because they aren't built to capture emergent properties – which I think is more or less what this sentence is trying to convey.

P3 (Section 2) – I think the first half of the Background could be folded into the Introduction, with some adjustments for repetition. There's a shift at P4L95 to the "motivation" for the model – the conceptual model by Psuty (2008) – that could set up the Methods as the start of Section 3. Then there is another shift at p4L115 that reaches toward implications, which as a reader I wasn't ready for until I'd seen the model do its work. I think those last two paragraphs of Section 2 would sit well in the Discussion, and establish a far more interesting beachhead than the current paragraphs about the Bruun rule (see related comment below).

P8–9 – I like the narrative of Section 3.3 – but I think it belongs in the Results as 4.1, where it would nicely set up the subsequent sections that discuss trade-offs between variables.

P14, P15 – Glad to see the nod to alongshore sediment transport and vegetation feedbacks. Plenty of scope for further work – and the right acknowledgement here of what's important but just out of scope.

P12L370 – Does there need to be so much text chasing the Vousdoukas et al. (2020)

result? (I understand the rhetorical move to make the underlying point about nonlinear versus linear response to exogenous forcing.) Perhaps the second paragraph of that point – P13L387 – is what pushes it slightly out of balance. The idea of asking what it take for this model to return those rates of retreat is interesting, but it reads as a kind of excursion – all the more because it begins the Discussion. Rather than cut it completely, it could just be condensed.

I enjoyed reading this contribution, and I look forward to seeing it in its final form.

References

McNamara, D. E., & Werner, B. T. (2008a). Coupled barrier island–resort model: 1. Emergent instabilities induced by strong human‐landscape interactions. Journal of Geophysical Research: Earth Surface, 113(F1).

McNamara, D. E., & Werner, B. T. (2008). Coupled barrier island–resort model: 2. Tests and predictions along Ocean City and Assateague Island National Seashore, Maryland. Journal of Geophysical Research: Earth Surface, 113(F1).

Werner, B. T. (2003). Modeling landforms as self-organized, hierarchical dynamical systems. In Prediction in Geomorphology. Geophys. Monogr. Ser. 135, edited by P. R. Wilcock and R. M. Iverson, AGU, Washington, D.C., 133–150.

---

## Referee Comment (RC2) · Anonymous Referee #2 · 18 Dec 2020

The article is well written. I appreciate the richness of results resulting from very simple equations. I think this is a good model to think about real settings. I don't have major issues with it. Instead, I have some minor comments / discussion.

1-I was confused about the parameter Dt throughout the article. If I understand correctly, this is both 1) the accommodation depth of the shoreface, 2) sandy substrate thickness, and 3) the inner profile closure depth (line 510). Can you better describe this parameter and all its interpretations early in the paper? More importantly, how does it relate to the classic depth of closure (which for century time scales should be much larger than 5 m, and much larger than 2 m).

[Figure]

2-Would you be able to make a comparison between your model and the model of LTA14? Is there anything that your model can do while the LTA14 can't? Can the two models be easily merged, or do they use incompatible schematizations?

3-The authors found very rapid behavioral changes triggered by small changes in parameters (e.g., SLR>5 mm/yr). Even though this is plausible, I encourage the authors to consider a limitation of their model. Their model arbitrarily and independently fixes the fluxes Qs and Qd. As a result, the model does not have many degrees of freedom. The analogy is trying to simulate hydrodynamics by imposing boundary conditions very close to the area of interest: there is not much room for smoothing them and the system has a very stiff response. In reality, the fluxes Qs and Qd should not be fixed. For example, the foredune flux should decrease when dunes are larger. Also, Qs and Qd might not be completely independent. For example, larger waves might increase both Qs and Qd. Could you comment on these feedbacks?

4-The color scheme is confusing. It goes to dark to white to black. In Fig. 8 bottom-left it seems that there are sharp discontinuities in the behavior (i.e., the horizontal streaks for Qs>30). But I think this is an artifact of the color scheme. (instead, I think that there are parts of the plot where discontinuities are real, e.g., between blue and yellow). You can check out scientific appropriate color schemes here https://www.nature.com/articles/s41467-020-19160-7

5- Fig 3,4,5,. What is the slope of the backbarrier? Is it a parameter that affects the model result? Or is it just a graphical add-on? Please specify.

Line 404. Not a good form to start a paragraph with however Line 451. Suggests that our model

---

## Author Comment (AC1) · 31 Jan 2021

[[ This work by Ciarletta & colleagues is an engaging and well-written exploration of cross-shore barrier dynamics in a deliberately simplified numerical model. The authors are clear about what the model does and does not explicitly address, and that it serves a tool for first-order quantitative insight into transitions between barrier states and behaviors otherwise framed in conceptual terms. The differentiated regime space indicates the rich dynamics that the model is capable of simulating, and the authors are careful to keep one foot in the real world by constraining the fundamental parameters of the model with empirical rates. ]]

[Figure]

+ Thank you. We hope to deliver additional first-order insights through field-model comparison in the near future.

[[ My comments are minor, but I hope offer some helpful suggestions:

Abstract, Introduction – To me, snippets of the Abstract (and related snippets of the Introduction) are written in a way that suggests a decade is approximately the upper end of the model-world time scale here. (Other parts of the manuscript are clearer in this regard.) Just to reflect: the model runs on an annual increment and can comfortably tick over for a couple of centuries, which means the interesting changes occur on decadal time scales. If the authors describe it as a model of multi-decadal barrier dynamics – on the order of 101–102 years – I think that might help readers have a better sense from the outset of what the model does and doesn't do. (And it creates useful space for the interesting discussion, late in the manuscript, regarding what explicitly including event-driven changes and fast-acting processes might add.) ]]

+ We agree that the way we currently describe this is a bit confusing, and stating as "multi-decadal" instead of "decadal" seems like a straightforward way to make this clearer.

[[ P3L66 and elsewhere – Recommend cutting "we believe," since it's implicit in the conditional statement that follows. Would also change instances of "believe" on P14L440/442 – prefer "thought to be..." or similar. ]]

+ Recommendations implemented.

[[ P3L76 – "Whole-barrier dynamics at the mesoscale (10s to 100s yrs) are poorly represented by models, partly because the complexity of geomorphic processes at this scale cannot be easily represented by simple linear relationships..." – I think Werner (2003) on complex hierarchy offers a helpful explanation here (because what manifests at the mesoscale is a mix of linear and nonlinear), and I'd offer that McNamara & Werner (2008a, 2008b) are one example of a barrier model that does quite well in

this regard (and ultimately manages to account for both natural and human-dominated conditions). The operational blocks of this model (i.e., triangles, partitioned distribution) in many ways appear related, at least conceptually, to that earlier work. And even if the similarity ends there, it's a sound foundation. Furthermore, the spatially extended version of the LTA14 model exhibits some interesting mesoscale behaviour – which the authors cite (Ashton & Lorenzo-Trueba, 2018). All that is to say – I suggest the authors cut back the "poorly understood" spin throughout the Introduction and take a lighter, more open tone like the one at P3L80 – a framing of the problem that makes plenty of room for previous work. Another reason for that subtle adjustment is that this work, too, is, as the authors rightly state, a "first-order" model, steered but hardly constrained by empirical parameters.

Sticking with this same introductory sentence – I'm not sure that geomorphic processes operate at the whole-barrier scale (and I think the authors would concur). Returning to Werner (2003) – whole-barrier dynamics are an emergent property of geomorphic processes of sediment transport at smaller-than-whole-barrier scales. So perhaps the authors could revisit their warrant here. My sense is that models struggle to represent whole-barrier dynamics because they aren't built to capture emergent properties – which I think is more or less what this sentence is trying to convey.

P3 (Section 2) – I think the first half of the Background could be folded into the Introduction, with some adjustments for repetition. There's a shift at P4L95 to the "motivation" for the model – the conceptual model by Psuty (2008) – that could set up the Methods as the start of Section 3. Then there is another shift at p4L115 that reaches toward implications, which as a reader I wasn't ready for until I'd seen the model do its work. I think those last two paragraphs of Section 2 would sit well in the Discussion, and establish a far more interesting beachhead than the current paragraphs about the Bruun rule (see related comment below). ]]

+ We will address the aforementioned points together, since they are interrelated. After consideration for the response from another reviewer, we agree that the first paragraph

of the Background is largely redundant with the Introduction and work this into the third paragraph of Section 1 (see modifications below). We also agree with the suggestions to cut back on the "poorly represented" language and add citations for Werner (2003) and McNamara and Werner (2008ab). We integrate these references into the Introduction to clarify the philosophy behind the model (chiefly, the characteristic timescale of barrier landscape formation), as well as provide some context for what such a model is built to explore.

We retain the remaining Background section with some light modifications (particularly the last two paragraphs) since this is information that helps explain the need for the model, as well as define how some of the model parameters (e.g. vertical accommodation) are understood from the field. The latter point is relevant since another reviewer specifically requested that we clarify early in the paper what vertical accommodation is and how it varies from one location to another.

To be clear, we feel it is important to make the distinction early on that this framework is not simply a rehash of the earlier model (Ciarletta et al., 2019), but a greatly expanded system specifically designed to answer questions that go beyond Psuty's conceptual model and exceed the capabilities of the old framework. These questions require some background to understand, especially for researchers who may be working in seemingly disparate environments, and we attempt not to blindside in the Discussion.

Modifications to Introduction, Paragraph 3 (P2): "[. . .] However, such a model need not be event-based to approximate the net result of flux-driven changes in time, allowing for reduced-complexity simulation (French et al., 2016). This is consistent with the hierarchal view of natural systems by Werner et al. (2003), which considers characteristic timescales of landscape self-organization from processes occurring over shorter time intervals. The concept is particularly suited to mesoscale modeling of barrier evolution, where emergent morphology is at least partly understood from observational and historical records (Psuty, 2008; Psuty and Silveira, 2013) but cannot be easily driven by linear relationships alone (Cooper et al., 2018). Furthermore, by idealizing the geometry of barrier systems, it is possible to partition sediment volume within a simple deterministic framework, relying on geometric and algebraic relationships to shape the morphology of the system as a function of not just sediment fluxes, but changes in other external forcing (e.g. changing accommodation due to SLR). A similar type of modeling has been accomplished by McNamara and Werner (2008a/b) who constructed a geometrically simplified barrier model with partitioned sediment distribution driven by beach replenishment tied to human development."

[[ P8–9 – I like the narrative of Section 3.3 – but I think it belongs in the Results as 4.1, where it would nicely set up the subsequent sections that discuss trade-offs between variables. ]]

+ We've renumbered this section and made some minor adjustments to wording to integrate as the first section of the results.

[[ P14, P15 – Glad to see the nod to alongshore sediment transport and vegetation feedbacks. Plenty of scope for further work – and the right acknowledgement here of what's important but just out of scope. ]]

+ Thanks! We definitely tried to make sure the scope of our work is well understood. There is a lot of parallel research going on that could make for interesting collaborations and model integration down the road.

[[ P12L370 – Does there need to be so much text chasing the Vousdoukas et al. (2020) result? (I understand the rhetorical move to make the underlying point about nonlinear versus linear response to exogenous forcing.) Perhaps the second paragraph of that point – P13L387 – is what pushes it slightly out of balance. The idea of asking what it take for this model to return those rates of retreat is interesting, but it reads as a kind of excursion – all the more because it begins the Discussion. Rather than cut it completely, it could just be condensed. ]]

+ We agree that some of paragraph 13 is a bit excessive. To streamline it, we remove

most of the first part of this paragraph relating directly to Vousdoukas:

"Additional factors that may buffer the potential loss of natural beaches include preexisting dune volume and island width. Wide barriers, in particular, can provide space for subaerial accumulation and a glut of sediment to directly counter erosion caused by sand deficits and increasing SLR. This is exemplified by formerly and presently wide barrier islands such as Cedar and Parramore islands in Virginia–both of which were historically around four times wider than the 400 m wide barrier initialized in our model investigation. Despite experiencing an acceleration in relative SLR of 3 to 4 mm yr-1 over the last century (Boon and Mitchell, 2015), these islands have recently or historically sustained kilometer-scale landward shoreline migration over decadal to centennial timescales (McBride et al., 2015; Deaton et al., 2017; Shawler et al., 2019). Similar longer-term and sustained narrowing of previously wide barriers has also been inferred at the Bogue Banks, North Carolina, a system of formerly progradational islands that began to undergo net shoreline erosion approximately 1 kya (Timmons et al., 2010). The combination of our modeling results and observations from natural systems therefore suggest that net sand surpluses over geological to historical timescales that serve to enhance system volume storage may render barriers more resistant to periods of sediment deficit or accelerated SLR, particularly over the mesoscale."

---

## Author Comment (AC2) · 31 Jan 2021

[[ The article is well written. I appreciate the richness of results resulting from very simple equations. I think this is a good model to think about real settings. I don't have major issues with it. Instead, I have some minor comments / discussion. ]]

+ Thanks. We were inspired to construct and explore this framework by the work of many other researchers, and we hope the community finds utility in this model or its results.

[[ 1-I was confused about the parameter Dt throughout the article. If I understand

correctly, this is both 1) the accommodation depth of the shoreface, 2) sandy substrate thickness, and 3) the inner profile closure depth (line 510). Can you better describe this parameter and all its interpretations early in the paper? More importantly, how does it relate to the classic depth of closure (which for century time scales should be much larger than 5 m, and much larger than 2 m). ]]

+ We modify the last paragraph of the Background to better explain this, as Dt does have multiple controls depending on geologic context. In some cases, Dt is directly controlled by the presence of a consolidated sediment or bedrock interface, as it is in the Outer Hebrides and the Gulf Coast of Florida (allowing very small Dt). In other cases, where sediments are unconsolidated to depth, the accommodation available at the shoreface is based on the depth of the wave ravinement surface. We know from field observations that this depth is generally shallower than what would be expected of a classic depth of closure. For example, if we look at places like Fire Island (NY) and Parramore Island (VA) that have experienced progradation over the last centuries, we can see that there is 4 to 6 meters of sediment overlying what was geologically recently (centuries ago) seabed. As such, our best guess is that this vertical accommodation is more related to the inner depth of closure, which Hallermeier (1978) suggests as the seaward limit of shoreface that is significantly shaped by alongshore sediment transport processes (that we posit are responsible for most Qs fluxes in non-headland beach systems). Since changes in alongshore sediment fluxes occur significantly at sub-centennial timescales, translations of the uppermost subaqueous shoreface would likely occur over depths less than or equal to the inner depth of closure itself, and mostly independent of the outer depth of closure. Even where a barrier prograades significantly beyond the initial cross-shore location of the inner depth of closure over longer timescales, we believe the vertical accommodation to be filled does not change significantly over the spatial scales consistent with progradation (kilometers or less). This is because the slope of the shoreface in real-world systems becomes flatter with increasing offshore distance.

Finally, we consider that even the classic depth of closure could be very small in fetch-limited environments like bays and large lakes. In these cases, the inner depth of closure would be correspondingly small, and could help explain how barriers in places like the Great Lakes appear very dynamic despite limited energy availability.

Modifications to last paragraph of Background: "Moreover, sandy-substructure accommodation (the vertical space needed to be filled or eroded to invoke shoreline migration over multi-decadal scales) differs across the globe due to both local geology and available wave energy. In some cases, vertical accommodation is solely a function of antecedent geology, where consolidated sediment and bedrock interfaces define the seaward transgressive surface of the shoreface. In other systems with unconsolidated sediments, the depth of the shoreface available to be filled is a more a function of wave climate and uppermost shoreface lithology. Combinations of these influences are possible, which suggests the baseline sensitivity of barriers to sediment input/loss magnitudes varies considerably."

[[ 2-Would you be able to make a comparison between your model and the model of LTA14? Is there anything that your model can do while the LTA14 can't? Can the two models be easily merged, or do they use incompatible schematizations? ]]

+ The biggest differences between the SBSP model and LTA14 are that the latter has a parameterized shoreface and consideration for backbarrier lagoon depth, while the former has relatively detailed subaerial morphology and rudimentary stratigraphic capability. The schematizations are only partly incompatible, and it may be possible to merge these two by using LTA14's shoreface to drive the direction and magnitude of Qs. Additionally, the merged model could incorporate some aspect of LTA14's overwash component to fill the backbarrier and control fluxes to the lagoon. It is not perfectly clear how this would work, since there are considerations for how overwash impacts any existing topography in a cross-shore profile. Furthermore, recent field data seems to suggest that storm-driven overwash events are more complicated than depicted in LTA14, with sediment movement both onshore and offshore from the subaerial system.

[[ 3-The authors found very rapid behavioral changes triggered by small changes in parameters (e.g., SLR>5 mm/yr). Even though this is plausible, I encourage the authors to consider a limitation of their model. Their model arbitrarily and independently fixes the fluxes Qs and Qd. As a result, the model does not have many degrees of freedom. The analogy is trying to simulate hydrodynamics by imposing boundary conditions very close to the area of interest: there is not much room for smoothing them and the system has a very stiff response. In reality, the fluxes Qs and Qd should not be fixed. For example, the foredune flux should decrease when dunes are larger. Also, Qs and Qd might not be completely independent. For example, larger waves might increase both Qs and Qd. Could you comment on these feedbacks? ]]

+ The motivation behind this model is to test the relationship of these fluxes to morphology at the most basic level, and based on this comment, we consider that it has provoked precisely the type of thought that it was intended to encourage. We acknowledge here that we are mostly testing the magnitude difference between the subaerial and subaqueous fluxes, and so behavioral boundaries are understandably rigid. That being said, we can speculate on what is actually happening in natural systems. As we point out in the discussion, one of the major forces potentially driving real-world systems is deflation (Qw), which itself is probably modified by time-variable controls such as climate and vegetation. Even if Qd was somehow fairly static over decadal timescales, the inclusion of Qw competing with it in the subaerial domain would almost certainly reshape our regime plots to some extent, and could result in true equilibria for dune volumes (e.g. where dune Qd and Qw are balanced with respect to volume losses to sea-level rise). Additionally, while our model is somewhat rigid as currently parameterized, it is worth mentioning that our framework does pick up on the slowing of dune growth caused by Qd being distributed over a larger dune profile with increasing time (see section 3.3, paragraph 2)—a concept recently discussed by Davidson-Arnott et al. (2018).

Qs and Qd are also certainly related to each other, as wave energy shapes the

sedimentology of the system itself, and waves are dependent on wind. Accordingly, Jackson et al. (2019) points out that coastal erosion and the development of large/transgressive dunes likely occurred synchronously in response to increasing windiness during the Little Ice Age. As discussed in section 5.2, what we find intriguing is how the magnitudes of Qs and Qd change with respect to each other under different energy regimes. If the magnitude of Qd increases faster than Qs with increasing system energy (windiness), this might explain how barriers undergo transitions to/from dune-dominated morphologies.

[[ 4-The color scheme is confusing. It goes to dark to white to black. In Fig. 8 bottom-left it seems that there are sharp discontinuities in the behavior (i.e., the horizontal streaks for Qs>30). But I think this is an artifact of the color scheme. (instead, I think that there are parts of the plot where discontinuities are real, e.g., between blue and yellow). You can check out scientific appropriate color schemes here https://www.nature.com/articles/s41467-020-19160-7 ]]

+ This is not an artifact, but it certainly could appear that way. Once the barrier begins to undergo sustained progradation (Qs>30), it is not just the height of ridges that changes at the end of each 500-year simulation, but also the number of ridges, which is in some cases affected by amalgamation. Compare the bottom left and right plots of Figure 8. The discontinuities in active ridge height generally line up with the number of ridge crests produced, but not always. Where these plots do not align, this is because amalgamation can reduce the number of ridges while maintaining the height of the active ridge.

The overall scheme for these figures was constructed specifically to be readable to persons with color deficiencies (test on Coblis Colorblind Simulator https://www.color-blindness.com/coblis-color-blindness-simulator/), as well as highlight important trends in the output. However, we acknowledge the data is genuinely hard to interpret because of the reasons stated above. We add this discussion to the caption of Figure 8 to help dispel any confusion:

"Note that the active ridge height and number of ridge crests do not change synchronously at the end of each 500-year simulation due to the presence of amalgamation. Discontinuities in the plot of active ridge height generally align with the plot of ridge crests produced, but where they differ it is because amalgamation can reduce the number of ridges while maintaining the height of the active ridge."

[[ 5- Fig 3,4,5,. What is the slope of the backbarrier? Is it a parameter that affects the model result? Or is it just a graphical add-on? Please specify. ]]

+ The slope of the underlying sandy platform is a graphical feature. We will add a line to the captions to specify this: "The backbarrier slope of the sandy platform is shown for illustrative purposes and is not currently parameterized in the model."

[[ Line 404. Not a good form to start a paragraph with however ]]

+ Agreed. "In contrast to natural systems," would be more appropriate. Edit made.

[[ Line 451. Suggests that our model ]]

+ Thank you. Insertion made.